# The chromatin remodeling factor ISW-1 integrates organismal responses against nuclear and mitochondrial stress

Olli Matilainen[1,3], Maroun S. Bou Sleiman[1], Pedro M. Quiros [1], Susana M.D.A. Garcia[2] & Johan Auwerx[1]

Age-associated changes in chromatin structure have a major impact on organismal longevity. Despite being a central part of the ageing process, the organismal responses to the changes in chromatin organization remain unclear. Here we show that moderate disturbance of histone balance during *C. elegans* development alters histone levels and triggers a stress response associated with increased expression of cytosolic small heat-shock proteins. This stress response is dependent on the transcription factor, HSF-1, and the chromatin remodeling factor, ISW-1. In addition, we show that mitochondrial stress during developmental stages also modulates histone levels, thereby activating a cytosolic stress response similar to that caused by changes in histone balance. These data indicate that histone and mitochondrial perturbations are both monitored through chromatin remodeling and involve the activation of a cytosolic response that affects organismal longevity. HSF-1 and ISW-1 hence emerge as a central mediator of this multi-compartment proteostatic response regulating longevity.

[1] Laboratory for Integrative and Systems Physiology, Institute of Bioengineering, École Polytechnique Fédérale de Lausanne, Station 15, 1015 Lausanne, Switzerland. [2] Institute of Biotechnology, University of Helsinki, FI-00014 Helsinki, Finland. [3] Present address: Institute of Biotechnology University of Helsinki, FI-00014 Helsinki, Finland. Correspondence and requests for materials should be addressed to J.A. (email: admin.auwerx@epfl.ch)

Organismal gene expression in response to stress is regulated through multiple mechanisms[1]. One of these mechanisms is modulation of chromatin structure, which plays an essential role in cellular stress tolerance. Therefore, epigenetic modulators have emerged as nodal points in stress responses and longevity. For example, mitochondrial stress in *Caenorhabditis elegans* promotes global chromatin silencing through H3K9me2 methylation[2]. In parallel to the widespread silencing, chromatin loci that harbor stress response genes are opened to allow the binding of transcription factors DVE-1 and ATFS-1 to initiate the mitochondrial unfolded protein response (UPR[mt])[2]. Along with global histone methylation[2], the histone H3K27me3 and H3K27me2 demethylases, JMJD-3.1 and JMJD-1.2, are required for specific H3K27 demethylation and activation of UPR[mt] genes upon mitochondrial stress[3]. Importantly, both epigenetic mechanisms, H3K9me2 methylation and H3K27me2/3

demethylation, are prerequisite for mitochondria-mediated extension of lifespan[2,3], highlighting an essential crosstalk between mitochondria and the epigenome[4]. In addition to its role in the UPR[mt], JMJD-3.1-mediated H3K27me3 demethylation maintains the heat-shock response after the onset of reproduction[5], indicating that epigenetic modifications are widely involved in cellular stress responses. Importantly, the epigenome itself is affected by ageing, which causes widespread changes in chromatin structure through alterations in processes, such as DNA methylation, histone modifications, chromatin remodeling, and heterochromatin packaging[6]. In addition to changes in epigenetic modifications, ageing is associated with the loss of histone proteins[7–9]. These age-associated epigenetic changes have a central role in the progression of ageing, as they contribute to genomic instability, mitochondrial dysfunction, and loss of proteostasis[6]. While the role of epigenetic modifications in stress

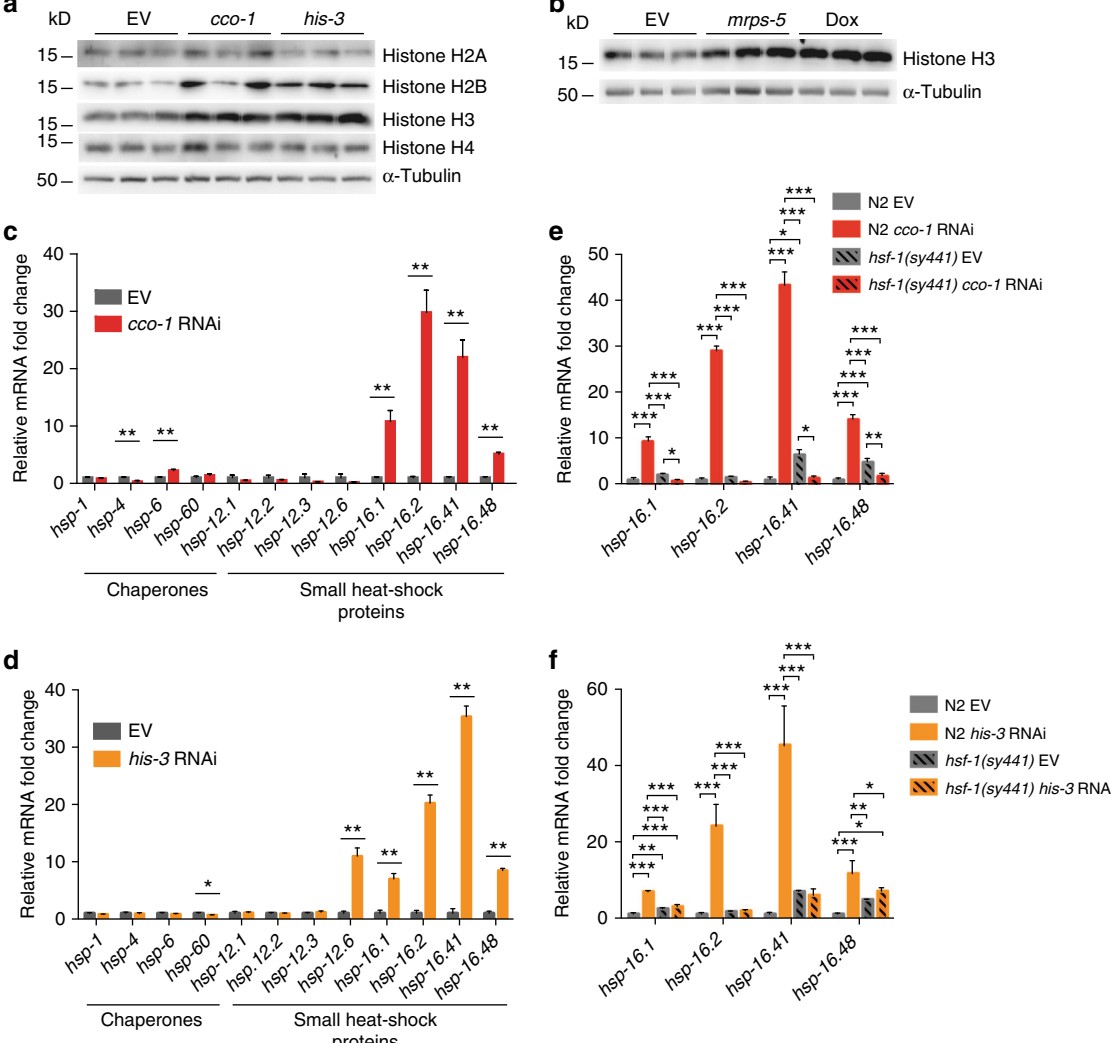

**Fig. 1** Mitochondrial and histone stress modulate histone levels and trigger sHSP response. **a** Histone western blots from N2 worms after treatment with *cco-1* or *his-3* RNAi. Results shown are from three technical replicates. Similar results were obtained in two to three independent experiments. **b** Histone H3 western blots from N2 worms after treatment with *mrps-5* RNAi or doxycycline (Dox). Results shown are from three technical replicates. Similar results were obtained in two to three independent experiments. **c, d** Chaperone expression in N2 worms after treatment with **c** *cco-1* or **d** *his-3* RNAi. In **c** and **d**, N2 worms were collected for qRT-PCR analysis at L4 stage. Bars represent mRNA levels relative to empty vector with error bars indicating mean ± s.d. of three biological replicates, each with three technical replicates (*$P < 0.05$, **$P < 0.01$; unpaired Student's *t*-test). **e, f** sHSP expression in N2 and *hsf-1(sy441)* mutant worms after treatment with **e** *cco-1* or **f** *his-3* RNAi. In **e** and **f**, worms were collected for qRT-PCR analysis at L4 stage. Bars represent mRNA levels relative to N2 worms treated with empty vector with error bars indicating mean ± s.d. of three biological replicates, each with three technical replicates (*$P < 0.05$, **$P < 0.01$, ***$P < 0.001$; one-way ANOVA with Tukey's post test)

responses and longevity is beginning to unravel, it is unknown how cells and organisms respond against changes in the epigenome.

Here, by using *C. elegans*, we show that perturbation of the histone balance activates a cytosolic stress response manifested by the induction of small heat-shock proteins (sHSPs). This stress response is not only dependent on HSF-1 but also on the chromatin remodeling factor ISW-1. Moreover, ISW-1 is required and sufficient to activate the cytosolic chaperone-mediated stress response, which leads to extended lifespan. Remarkably, also mitochondrial perturbations cause changes in histone levels, thereby activating the same HSF-1- and ISW-1-dependent cytosolic stress response. These data demonstrate that nucleus and mitochondria share a common stress response, which enhances the cytosolic proteostasis.

## Results

**Histone and mitochondrial stress modulate histone levels.** As increased histone expression extends lifespan in yeast[7], we asked which cellular processes are modulating histone levels within the organism. Interestingly, it has been shown that decreased histone expression and deficient nucleosome assembly increases mitochondrial DNA copy number, oxygen consumption, ATP synthesis, and oxidative phosphorylation in yeast[10]. However, it is not clear whether mitochondrial perturbation affects histone expression. To examine this, we treated *C. elegans* with RNAi against *cco-1*, a subunit of mitochondrial respiratory chain complex IV, and examined histone expression by quantitative reverse transcriptase-PCR (qRT-PCR). *cco-1* RNAi did not have a major effect on core histone expression, although it slightly increased the expression of the H3 genes (Supplementary Fig. 1a). In addition to histone H3 genes, *cco-1* RNAi increased the expression of *htz-1* (ortholog of mammalian H2A.Z) and the H1 linker histone gene *hil-1*, whereas the H1 linker histone gene, *his-24*, was downregulated (Supplementary Fig. 1a). In parallel with mitochondrial stress, we also asked how moderate histone stress via *his-3* RNAi, which targets nine different histone H2A genes, affects histone expression. Unlike many RNA interference (RNAi) clones targeting histones, *his-3* RNAi was selected for experiments, as it did not cause any visible developmental defects (data not shown). Treatment with *his-3* RNAi increased the expression of all measured histone genes except that of *his-24* (Supplementary Fig. 1a), suggesting that decreased expression of a subset of histone genes leads to compensatory effects. Although *cco-1* RNAi resulted in milder changes in histone expression compared to *his-3* RNAi (Supplementary Fig. 1a), western blot analysis showed that both treatments affect histone protein levels in a similar manner. *cco-1* RNAi increases the protein levels of histone H2B and H3, whereas the levels of histone H2A and H4 remain unchanged (Fig. 1a). Similarly, *his-3* RNAi elevated histone H2B and H3 protein levels but did not affect histone H4 and slightly decreased the H2A protein level (Fig. 1a). By histone H3 immunoblotting, we found that, in addition to *cco-1* knockdown, also RNAi against *mrps-5*, a mitochondrial ribosomal protein, and treatment with doxycycline, an antibiotic that interferes with mitochondrial translation[11,12], increases histone H3 content (Fig. 1b), hence further supporting the observation that mitochondrial perturbation modulates histone abundance. Altogether, these results demonstrate hence that both mitochondrial stress and perturbation of histone balance modulate histone protein levels. These data also indicates that alterations in histone abundance are not only due to changes in transcription, as both *cco-1* and *his-3* RNAi affect histone protein levels in a similar manner, although histone transcript levels differ between these two conditions.

**Histone and mitochondrial stress activate HSF-1 response.** As both *cco-1* and *his-3* RNAi modulate histone gene expression and protein levels (Fig. 1a and Supplementary Fig. 1a), we asked whether they have similar effects on global gene expression. To examine this, we performed mRNA sequencing with *cco-1* and *his-3* RNAi-treated worms. Even though *cco-1* RNAi resulted in more pronounced changes in gene expression, both RNAi treatments share differentially expressed genes when compared to EV control, with four times more shared induced genes than repressed ones (199 up vs. 51 down, Supplementary Fig. 1b). To examine which cellular processes are affected by the two RNAi treatments, we performed gene set enrichment analysis (GSEA) using gene ontology (GO) terms as well as manually curated gene sets that contain the chromatin remodelers and heat-shock proteins (Supplementary Fig. 1c and Supplementary Table 1). As expected, the GSEA analyses for cellular component GO terms showed that *cco-1* RNAi feeding resulted in the enrichment of mitochondrial terms, while *his-3* RNAi leads to the upregulation of genes associated with the term chromosome. When molecular function was analyzed, however, some common terms relating to nucleic acid binding emerged, indicating a certain convergence in the response to the two RNAi treatments despite the clearly different subcellular compartmentalization of the targeted gene products. Importantly, when we examined the affected biological process GO terms, the upregulated genes with both RNAis were enriched for immune- and stress-related terms. This is consistent with the fact that mitochondrial stress has been shown to activate the innate immune response in both *C. elegans* and mouse[13,14], thereby demonstrating that regulation of this response is common between mitochondria and histone stress.

Since *cco-1* RNAi is known to activate the proteostatic UPR[mt] response[15], we investigated whether mitochondrial and histone perturbations trigger some common responses to maintain protein homeostasis. Histone depletion and loss of nucleosomes lead to variability in gene expression, or transcriptional noise, and to global increases in gene expression[8,16]. In *Arabidopsis*, the molecular chaperone Hsp90 buffers transcriptional noise deriving from environmental stress to ensure normal development[17]. Furthermore, as Hsp90 is responsible for the nongenetic variability as an adaptation to stress in yeast[18], we hypothesized that changes in histone levels, like those observed upon *cco-1* and *his-3* RNAi, induce a chaperone-mediated stress response in order to meet the increased demands for proteome maintenance. Notably, in addition to their role in proteostasis, heat-shock proteins are also important regulators of innate immunity[19,20], the process commonly upregulated upon both *cco-1* and *his-3* RNAi (Supplementary Fig. 1c). GSEA revealed that heat-shock proteins are among the most enriched in upregulated genes after both *cco-1* and *his-3* RNAi (Supplementary Fig. 1c), indicating that both stressors activate proteostatic response. Next we measured the expression of different chaperones after treatment with *cco-1* or *his-3* RNAi. qRT-PCR showed that both treatments commonly and selectively upregulate the expression of small heat-shock proteins (sHSPs) *hsp-12.6*, *hsp-16.1*, *hsp-16.2*, *hsp-16.41*, and *hsp-16.48* (Fig. 1c, d), indicating that changes in histone balance trigger a proteostatic response in the cytosol. Increased chaperone expression upon both stressors was strictly heat-shock factor 1 (HSF-1) dependent, as *cco-1* or *his-3* RNAi applied in *hsf-1(sy441)* mutants failed to upregulate the expression of sHSPs (Fig. 1e, f). Taken together, although many cellular responses to *cco-1* and *his-3* RNAi are divergent, the innate immune and sHSP response are activated by both of these stressors. As the innate immune response upon mitochondrial stress has been shown to be regulated by the transcription factor ATFS-1 in *C. elegans*[13], we focused our analysis on elucidating the mechanisms underlying the sHSP response.

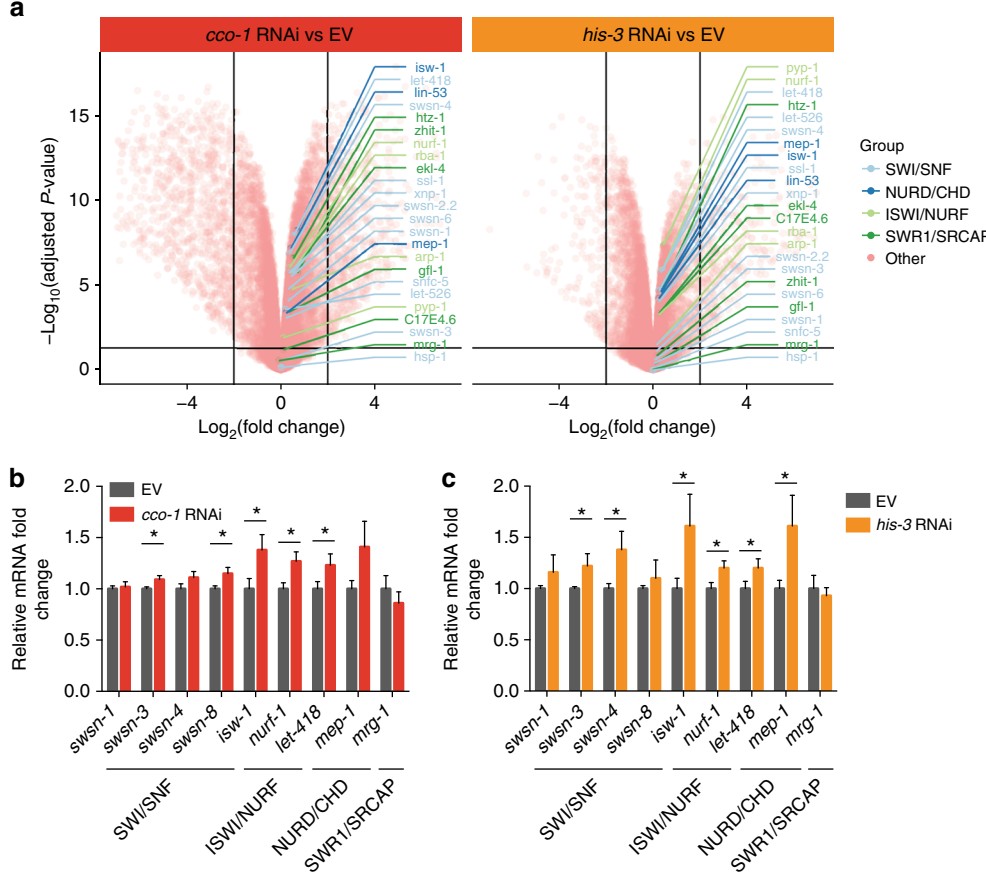

**Fig. 2** Mitochondrial and histone stress increase the expression of chromatin remodelers. **a** Volcano plots depicting changes in gene expression measured by RNA sequencing after treatment with *cco-1* or *his-3* RNAi. Chromatin remodelers[24] are highlighted in the plots. The order of chromatin remodelers on the right is based on adjusted *P*-value. **b**, **c** The expression of chromatin remodelers measured by qRT-PCR after treatment with **b** *cco-1* or **c** *his-3* RNAi. In **b** and **c**, N2 worms were collected for qRT-PCR analysis at L4 stage. Bars represent mRNA levels relative to empty vector with error bars indicating mean ± s. d. of three biological replicates, each with three technical replicates (*$P < 0.05$, **$P < 0.01$; unpaired Student's *t*-test)

**Histone and mitochondrial stress induce chromatin remodelers**. Since chromatin regulators have been shown to be important mediators of stress responses in yeast[21], we asked whether they also contribute to the sHSP response arising from mitochondrial (*cco-1* RNAi) and histone (*his-3* RNAi) stress, especially given their role in the yeast heat-shock response[22]. Chromatin remodelers are enzymes affecting chromatin among other things by positioning nucleosomes, inserting histone variants, and regulating nuclear organization[23]. Analyzing our RNA-seq data, *cco-1* and *his-3* RNAi were shown to increase the expression of different chromatin remodelers[24] (Fig. 2a, Supplementary Fig 1c, Supplementary table 1). Although the changes in chromatin remodeler expression are not among the strongest when looking at a global level, they are consistent between both treatments (Fig. 2a). Additionally, we selected members of SWI/SNF, ISWI/NURF, NURD/CHD, and SWR1/SRCAP chromatin remodeler complexes and examined their expression with qRT-PCR. Like seen in the RNA-seq experiment, also by qRT-PCR *cco-1* and *his-3* RNAi increased the expression of multiple chromatin remodelers (Fig. 2b, c). Again, although the changes in chromatin remodeler expression are not very strong, they are reproducible between independent experiments. These data suggest that histone and mitochondrial stress, which both affect histone levels, induce a chromatin remodeling response.

As changes in chromatin structure are an integral part of ageing[6], we asked whether chromatin remodelers upregulated upon histone depletion are involved in the regulation of longevity.

To test this, we examined how knockdown of *swsn-3*, *isw-1*, *nurf-1*, and *mep-1* affects *C. elegans* lifespan. Since chromatin remodeling enzymes have an important role during the development[25], we applied an RNAi regimen that ensures efficient gene knockdown already during embryogenesis and early larval stages. For this purpose, we initiated RNAi treatment within the L1–L3 stage of the previous generation (P0) and measured the lifespan of F1 generation (F1 generation still kept on RNAi plates). By using this "pretreatment" RNAi regimen i (Fig. 3a), we found that RNAi against *mep-1* and *nurf-1*, members of the NURD/CHD and ISWI/NURF complexes, respectively, extend lifespan and that RNAi against *isw-1*, a component of the ISWI/NURF complex, significantly reduces lifespan (Fig. 3b, Supplementary Table 2). Owing to the strong and previously unreported effect on longevity, we hence examined the role of ISW-1 in stress responses and longevity.

**Different stressors modulate *isw-1* expression**. ISW-1 is an ATP-dependent chromatin remodeling factor that can function as a part of the NURF (Nucleosome Remodeling Factor) complex in regulation of cell fate[26]. ISW-1 has been shown to be expressed in most of the *C. elegans* tissues[26]. Accordingly, in our transcriptional reporter strain, which expresses *gfp* under the *isw-1* promoter, we detected the *isw-1* expression in multiple tissues, with the intestine showing the strongest signal (Supplementary Fig. 2a). To complement the chromatin remodeler expression

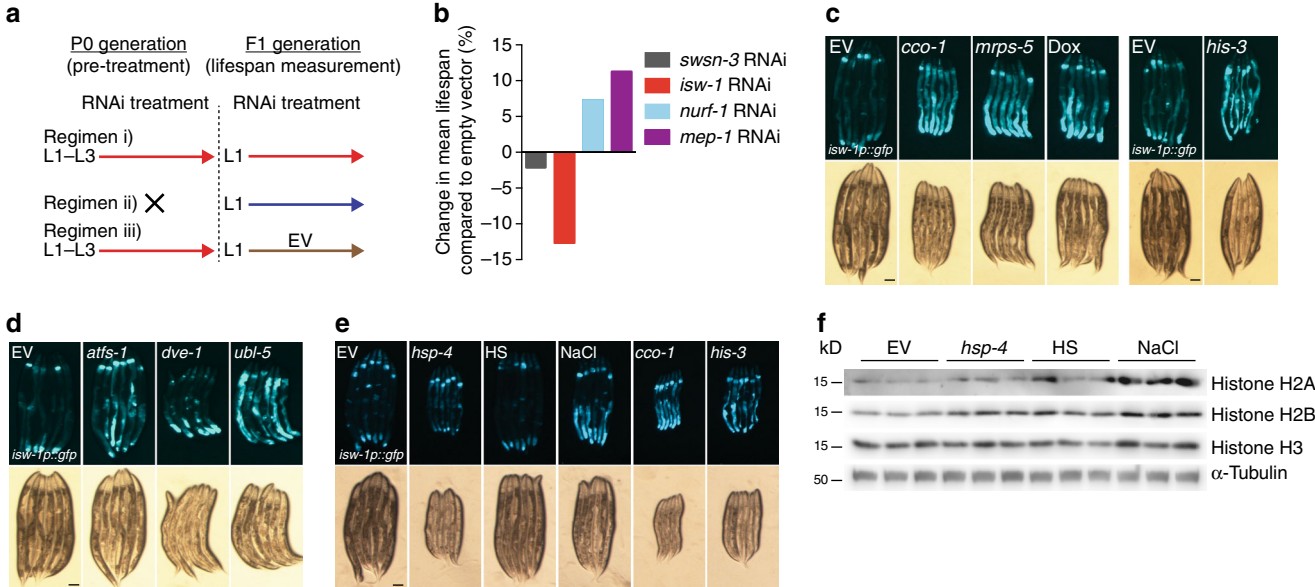

**Fig. 3** Stressors that modulate histone levels elevate *isw-1* expression. **a** Different RNAi regimens tested in lifespan experiments. (i) RNAi treatment initiated within L1–L3 larvae of previous generation (P0) used for experiment (F1) or (ii) within L1 larvae of generation used for experiment (F1). With regimen (iii), only P0 generation was treated with RNAi, and generation used for experiment (F1) was grown on plates with bacteria expressing empty vector. **b** Bar graph depicting N2 lifespan treated with RNAi against selected chromatin remodelers according to the RNAi regimen i. See Supplementary Table 2 for lifespan statistics. **c**–**e** *gfp* expression in *isw-1* transcriptional reporter (*isw-1p::gfp*) after **c** mitochondrial and histone stress, **d** RNAi of UPRmt core regulators and **e** ER−, heat or osmotic stress. Experiments with *isw-1p::gfp* reporter strain were repeated two to three times (*N* = 12–18). Scale bars, 100 μm. **f** The effect of ER−, heat or osmotic stress on histone H2A, H2B, and H3 protein levels. Results shown are from three biological replicates

data obtained by RNA-seq and qRT-PCR (Fig. 2a–c), we induced mitochondrial stress (by using *cco-1* RNAi, *mrps-5* RNAi, doxycycline) and histone stress (by using *his-3* RNAi) within the *isw-1* transcriptional reporter strain and analyzed the *gfp* expression. All treatments increased the green fluorescent protein (GFP) signal (Fig. 3c), demonstrating that both mitochondrial and histone stress have a robust effect on *isw-1* expression. Notably, RNAi against the UPRmt core regulators, *atfs-1*, *dve-1*, and *ubl-5*[27–29], also increased *isw-1* expression (Fig. 3d), suggesting that ISW-1 compensates for a defective UPRmt.

In addition to mitochondrial and histone stress, we investigated how endoplasmic reticulum (ER) stress (induced by feeding with a *hsp-4* RNAi), heat stress (in 37 °C), and osmotic stress (with NaCl) affect *isw-1* expression using the same transcriptional reporter strain. Heat stress and *hsp-4* RNAi did not affect or caused a mild increase in *gfp* expression, respectively (Fig. 3e). However, the increase in signal upon *hsp-4* RNAi was minor compared to the robust increase in *gfp* signal observed with *cco-1* and *his-3* RNAi (Fig. 3e). In contrast to ER− and heat stress, osmotic stress resulted in robust increase in the *gfp* signal (Fig. 3e). Next we examined how the above-mentioned stressors affect the levels of histone H2A, H2B, and H3, which were modulated upon *cco-1* and *his-3* RNAi (Fig. 1a). Similarly to the changes in *gfp* expression seen in transcriptional *isw-1* reporter (Fig. 3e), osmotic stress caused the most robust induction in histone levels (Fig. 3f). Taken together, these data indicate that *isw-1* expression increases in response to changes in histone levels induced by multiple stressors, although the effect on histones is not identical between different stress conditions (Figs 1a and 3f).

**ISW-1 regulates lifespan and sHSP-mediated stress response.** Next we examined the timing of ISW-1-mediated lifespan regulation. Whereas *isw-1* RNAi pretreatment robustly shortens lifespan (Fig. 3a, RNAi regimen i, Fig. 4a, Supplementary Table 2), *isw-1* RNAi treatment initiated from L1 larvae of the

generation used for lifespan experiment, i.e., in worms synchronized for lifespan by letting gravid hermaphrodites lay eggs for 4 h, did not affect lifespan (Fig. 3a, RNAi regimen ii, Fig. 4b, Supplementary Table 2). To examine whether *isw-1* RNAi only during the P0 generation affects the lifespan of the F1 generation, we grew the P0 generation on *isw-1* RNAi and let them lay eggs on plates with bacteria containing empty vector. We found that this maternal *isw-1* RNAi exposure (regimen iii) did not affect the lifespan of F1 generation (Fig. 3a, RNAi regimen iii, Fig. 4c, Supplementary Table 2).

Intriguingly, *isw-1* RNAi shortened the lifespan of F1 generation only when it was applied on both the P0 and F1 generation (Fig. 4a–c, Supplementary Table 2). Owing to this timing requirement, we hypothesized that, in order to reduce lifespan, *isw-1* expression has to be depleted during both early development and later in adulthood. Treatment of the P0 generation with *isw-1* RNAi strongly reduced the *isw-1* mRNA level in L1 larvae of F1 generation (Supplementary Fig. 2b), demonstrating that, within the lifespan-shortening RNAi regimen i (Fig. 3a), the F1 generation has low *isw-1* expression from early development onwards. When comparing *isw-1* expression between RNAi regimens i and ii (Fig. 3a) at the L4 stage of the F1 generation, we found that *isw-1* RNAi pretreated worms from RNAi regimen i had only slightly lower *isw-1* expression compared to worms from RNAi regimen ii without pretreatment (Supplementary Fig. 2c), suggesting that it is mainly the low *isw-1* expression during development that explains the short lifespan deriving from *isw-1* RNAi regimen i (Fig. 4a). Since *isw-1* RNAi regimen iii (Fig. 3a) did not shorten the lifespan of F1 generation (Fig. 4c), we hypothesized that, although *isw-1* RNAi of the P0 generation strongly decreases the *isw-1* expression in L1 larvae of the F1 generation (Supplementary Fig. 2b), growing these *isw-1* depleted L1 worms on bacteria-expressing empty vector restores the *isw-1* expression at later stages. To test this, we measured *isw-1* expression in day 2 adult worms treated with *isw-1* RNAi regimen i or iii. Despite the transgenerational stability of the

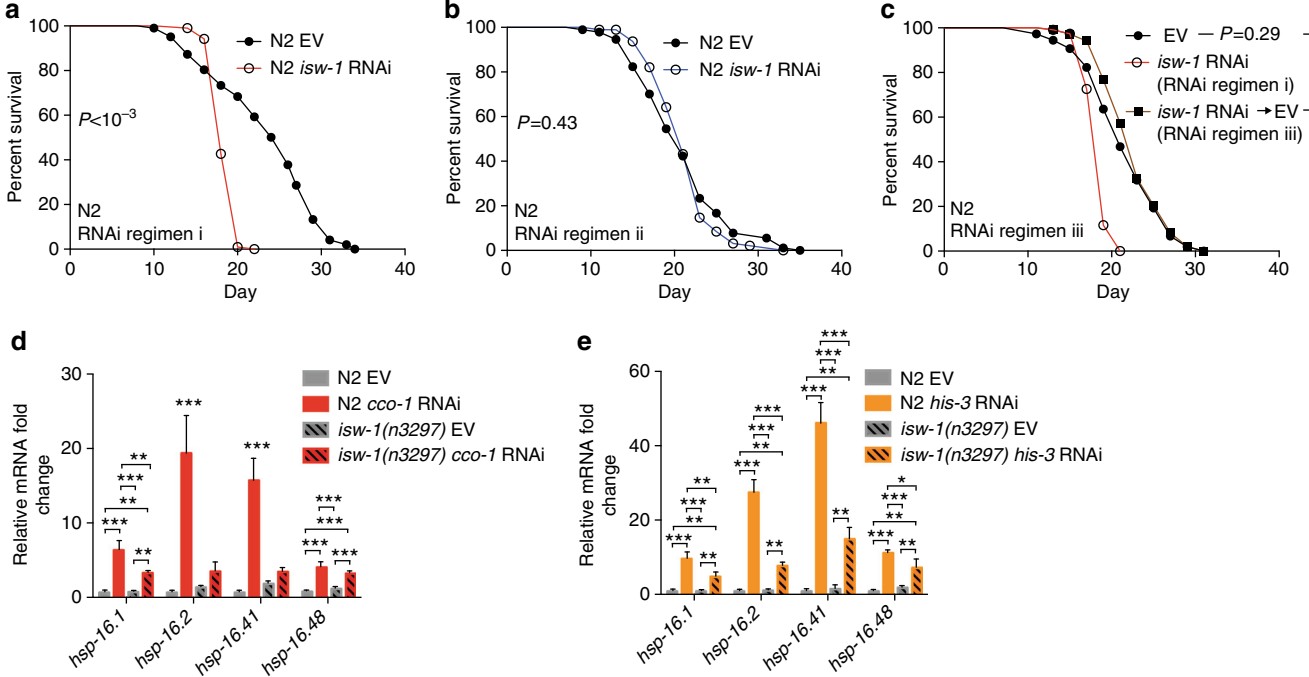

**Fig. 4** Reduced ISW-1 activity decreases lifespan and the expression of sHSPs. **a–c** N2 lifespan on *isw-1* RNAi according to RNAi regimen **a** i or **b** ii or **c** iii. See Supplementary Table 2 for lifespan statistics. **d**, **e** sHSP expression in N2 and *isw-1(n3297)* mutant worms upon **d** *cco-1* or **e** *his-3* RNAi. Worms were collected for qRT-PCR analysis at L4 stage. Bars represent mRNA levels relative to N2 worms treated with empty vector with error bars indicating mean ± s.d. of three biological replicates, each with three technical replicates (*$P < 0.05$, **$P < 0.01$, ***$P < 0.001$; one-way ANOVA with Tukey's post test)

RNAi effect in *C. elegans*, *isw-1* RNAi regimen iii allowed a partial recovery of *isw-1* expression in day 2 adult worms compared to the worms treated with RNAi regimen i (Supplementary Fig. 2d). This suggests that, even though maternal (P0 generation) *isw-1* RNAi reduced the *isw-1* expression during early development of the F1 generation (Supplementary Fig. 2b), partially restored *isw-1* expression at later stages (Supplementary Fig. 2d) rescues the precocious ageing phenotype and leads to normal lifespan (Fig. 4c). Collectively, these data demonstrate that, to shorten lifespan, *isw-1* expression has to be efficiently depleted during all life stages. Subsequently, as our data indicated that ISW-1 contributes to the longevity also at the later stages of life, we asked whether the abundance of ISW-1, and other chromatin remodelers[24], is altered in old *C. elegans* by re-analyzing an extant proteomic data set[30]. While the protein levels of many chromatin remodeling factors are slightly induced upon ageing, ISW-1 stands out as its expression is robustly increased in old worms (Supplementary Fig. 2e), hence supporting our hypothesis that ISW-1 regulates longevity also during the adulthood.

Next we examined whether ISW-1 is required for the increased sHSP expression upon *cco-1* and *his-3* RNAi treatments, and for this purpose, we used *isw-1(n3297)* mutants. Notably, the *isw-1(n3297)* mutation is predicted to cause one amino-acid substitution within a non-conserved region of the protein, leading to reduced ISW-1 activity but not to a complete loss of function[26]. The induction of sHSPs by *cco-1* and *his-3* RNAi treatment was blunted in *isw-1(n3297)* mutants when compared to wild-type worms (Fig. 4d, e), demonstrating that, in addition to HSF-1 (see Fig. 1e, f), ISW-1 mediates elevated sHSP expression upon mitochondrial and nucleosomal stress. Since both HSF-1 and ISW-1 are required for increased sHSP expression, we examined whether ISW-1 is affecting the binding of HSF-1 to the sHSP promoters. The four *hsp-16* sHSPs form two pairs that share promoters, each having two heat-shock response elements (HRE), which are HSF-1-binding sites (Supplementary Fig. 3a). We treated worms expressing GFP-tagged

HSF-1 with combined EV/*cco-1* or *isw-1*/*cco-1* RNAi to induce mitochondrial stress and worms expressing HA-tagged HSF-1 with combined EV/*his-3* or *isw-1*/*his-3* RNAi to induce histone stress and examined HSF-1 binding to sHSP HREs by chromatin immunoprecipitation followed by qRT-PCR. ISW-1 did not affect HSF-1 binding to these HREs (Supplementary Figs 3b, c), suggesting that ISW-1 regulates HSF-1 function through some other mechanism, or alternatively, it regulates sHSP expression independently of HSF-1.

To investigate whether the increased dosage of ISW-1 affects *C. elegans* stress responses and lifespan, we created a strain overexpressing *isw-1* under its endogenous promoter. In this *isw-1* overexpression strain (referred as *isw-1* OE strain from now on), *isw-1* expression was at its peak during the larval stages and thereafter rapidly decreased upon development (Fig. 5a). By examining the expression of different chaperone genes, we found that *hsp-16.1*, *hsp-16.2*, *hsp-16.41*, and *hsp-16.48*, which were induced upon *cco-1* and *his-3* RNAi (Fig. 1c, d), were likewise strongly upregulated in the *isw-1* OE strain (Fig. 5b). Elevated *isw-1* expression did not affect other proteome maintenance pathways, as proteasome activity was similar between wild-type and *isw-1* OE strain (Supplementary Fig. 4a), although qRT-PCR showed a slight tendency towards a decrease in proteasomal gene expression (Supplementary Fig. 4b). In addition, the expression of transcripts related to autophagy and mitophagy were unchanged in *isw-1* OE strain (Supplementary Fig. 4c). To examine whether changes in histone levels are mediated by ISW-1, or in other words, whether ISW-1 functions upstream of the histones, we tested whether *isw-1* OE affects histone H3 levels. Immunoblot analysis of Histone H3 levels showed that this was not affected in the *isw-1* OE strain (Supplementary Fig. 4d), thus indicating that the observed phenotypes in the *isw-1* OE strain are not due to the changes in histone levels.

Since whole-life *isw-1* knockdown shortens lifespan (Fig. 4a), we asked how *isw-1* overexpression affects lifespan. Despite the

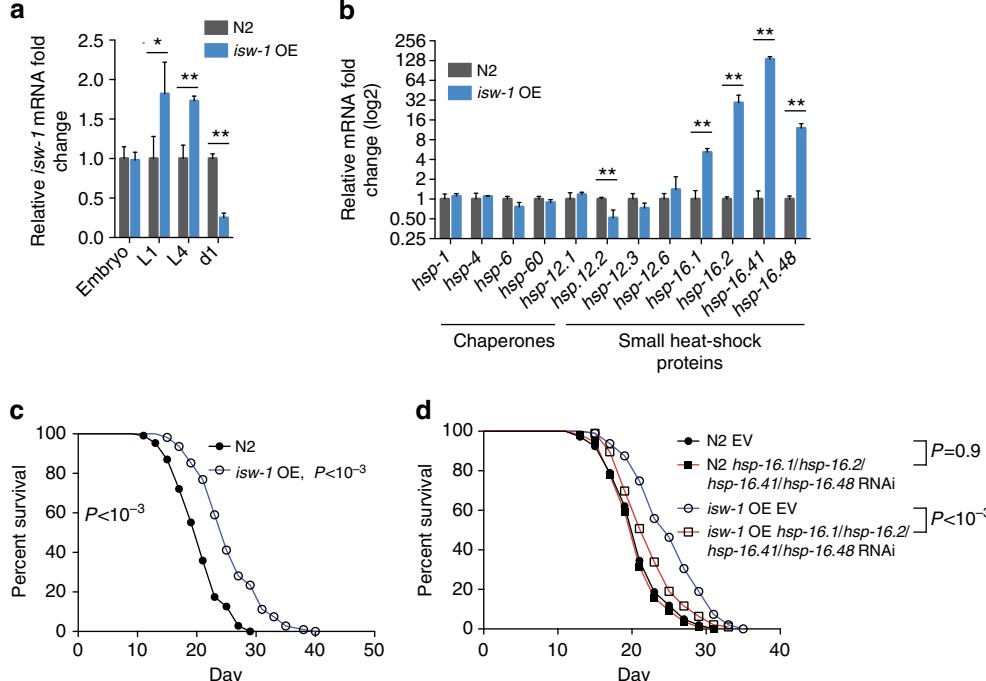

**Fig. 5** ISW-1 overexpression increases lifespan and the expression of sHSPs. **a** *isw-1* expression in *isw-1* OE strain. **b** Chaperone expression in *isw-1* OE strain. In **b**, worms were collected for qRT-PCR analysis at L4 stage. In **a** and **b**, bars represent mRNA levels relative to N2 worms treated with empty vector with error bars indicating mean ± s.d. of three biological replicates, each with three technical replicates (*$P < 0.05$, **$P < 0.01$; unpaired Student's *t*-test). **c** N2 and *isw-1* OE strain lifespan. **d** N2 and *isw-1* OE strain lifespan on sHSP RNAi. See Supplementary Table 2 for lifespan statistics

temporal nature of the elevated *isw-1* expression (Fig. 5a), we found that the *isw-1* OE strain lived significantly longer compared to the wild type (Fig. 5c, Supplementary Table 2). To confirm that longevity of *isw-1* OE strain is dependent on *isw-1*, we measured the lifespan of this strain on *isw-1* RNAi. Importantly, RNAi was done according to the regimen ii (Fig. 3a) to avoid the effect of *isw-1* RNAi on lifespan at normal conditions. *isw-1* RNAi abolished the lifespan extension of the *isw-1* OE strain (Supplementary Fig. 5a, Supplementary Table 2), thereby further highlighting the role of ISW-1 on longevity.

Chaperones are lifespan mediators of long-lived mutants[31], and therefore, we asked whether these stress proteins are contributing to *isw-1* OE strain longevity. In order to reduce the expression of each upregulated sHSP, we treated *isw-1* OE strain simultaneously with *hsp-16.1*, *hsp-16.2*, *hsp-16.41*, and *hsp-16.48* RNAi. This combined RNAi treatment did not affect the lifespan of wild type but significantly decreased the lifespan of *isw-1* OE strain (Fig. 5d, Supplementary Table 2), demonstrating that increased sHSP expression in *isw-1* OE strain is a prerequisite for its effect on longevity.

**ISW-1 regulates mitochondria- and histone-mediated longevity**. Owing to its role in driving the cytosolic sHSP response upon mitochondrial stress, we examined whether ISW-1 is contributing to the mitochondrial-mediated longevity. *cco-1* RNAi resulted in a much less pronounced lifespan increase in *isw-1*(*n3297*) mutants compared to wild-type worms (Fig. 6a, Supplementary Table 2). Notably, *cco-1* RNAi modulates histone levels also in *isw-1*(*n3297*) mutants, as both histone mRNA and histone H3 protein levels are altered (Supplementary Figs 5b, c), thereby demonstrating that reduced lifespan extension in *isw-1*(*n3297*) mutants upon mitochondrial perturbation (Fig. 6a) is not due to the lack of a modulating effect on histones.

To further study whether ISW-1 is required for mitochondria-mediated longevity, we treated worms simultaneously with *isw-1* and *cco-1* RNAi according to the RNAi regimen ii (Fig. 3a). Although ISW-1 has been shown to be required for RNAi function[32], simultaneous *isw-1* and *cco-1* RNAi treatment did not affect the *cco-1* knockdown efficiency (Supplementary Fig. 5d). Similarly, as observed with the *isw-1*(*n3297*) mutants (Fig. 6a), reduced *isw-1* expression did not completely suppress but significantly blunted the lifespan extension arising from *cco-1* knockdown (Fig. 6b, Supplementary Table 2). Collectively, these results demonstrate that ISW-1 is required for lifespan extension deriving from mitochondrial perturbation.

As histone stress was found to trigger sHSP and chromatin remodeler response (Figs 1d and 2a, c), we investigated whether it also affects longevity. *his-3* RNAi resulted in a slight but significant extension of lifespan in wild-type N2 worms (Fig. 6c, Supplementary Table 2). However, *his-3* RNAi failed to extend lifespan of *isw-1*(*n3297*) and *hsf-1*(*sy441*) mutants (Fig. 6d and Supplementary Fig. 5e, Supplementary Table 2), thus demonstrating that these factors, which are mediating the increase in sHSP expression upon *his-3* RNAi (Figs 1f and 4e), are also required for histone-mediated longevity. Although combined EV/*his-3* RNAi increased wild-type lifespan, combined treatment with *cco-1*/*his-3* RNAi did not further extend the long lifespan of EV/*cco-1* RNAi-treated worms (Fig. 6e, Supplementary Table 2), indicating that histone and mitochondrial stress extend lifespan through similar mechanisms.

**Discussion**
Chromatin remodeling contributes to stress responses and longevity, as, for example, the SWI/SNF complex is required for responses against heat and oxidative stress and for normal lifespan in *C. elegans*[33]. The observations that mild *isw-1*

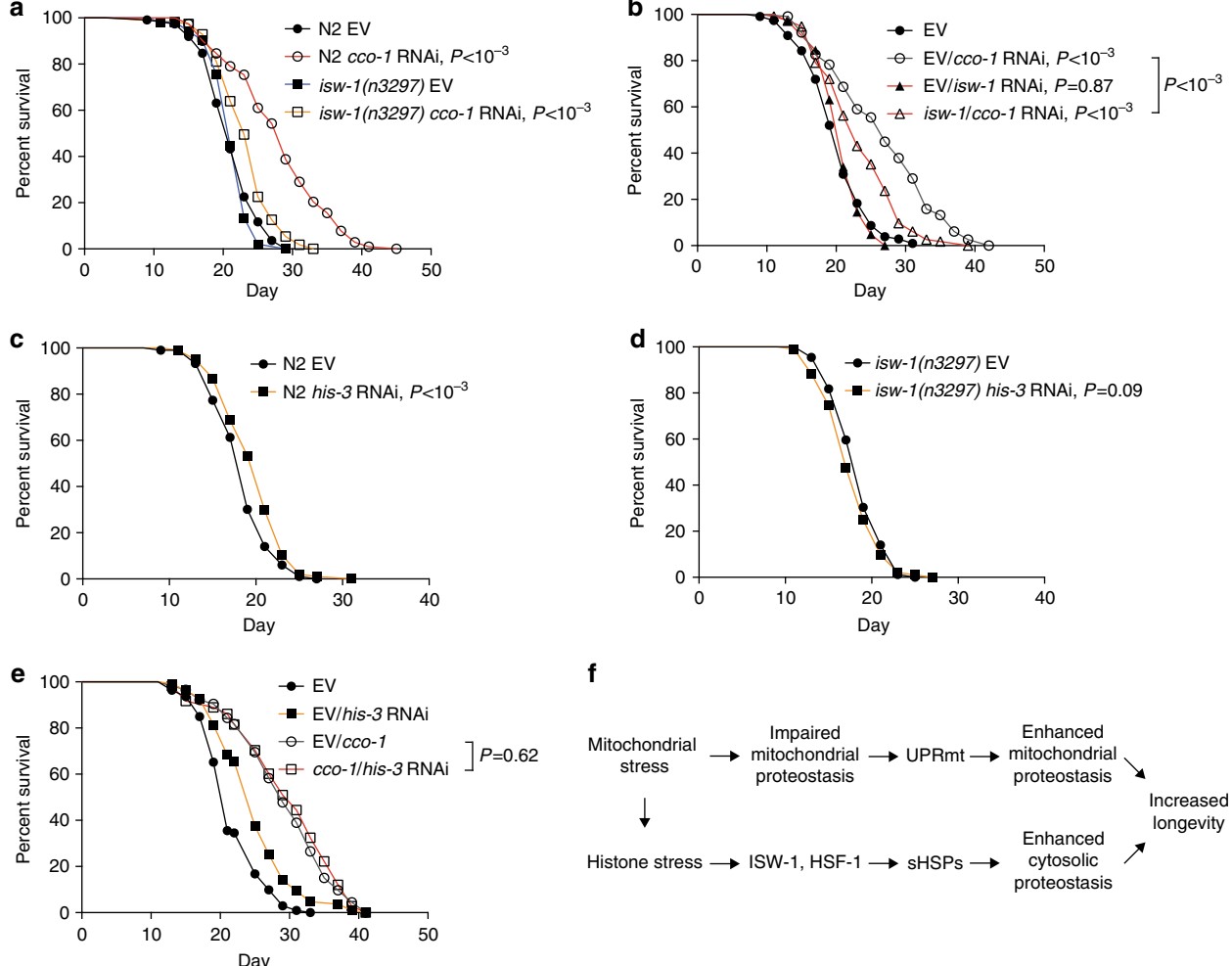

**Fig. 6** ISW-1 is required for mitochondria- and histone-mediated longevity. **a** N2 and *isw-1(n3297)* mutant lifespan on *cco-1* RNAi. **b** N2 lifespan on combined *isw-1* and *cco-1* RNAi according to RNAi regimen ii (see Fig. 3a). **c** N2 lifespan on *his-3* RNAi. **d** *isw-1(n3297)* mutant lifespan on *his-3* RNAi. **e** N2 lifespan on combined *cco-1* and *his-3* RNAi. See Supplementary Table 2 for lifespan statistics. **f** Model based on data presented here

overexpression is sufficient to robustly increase the expression of only sHSPs (Fig. 5b) and that the longevity of *isw-1* OE strain is significantly blunted by decreased sHSP expression (Fig. 5d) are strong indicators that ISW-1 controls specifically the expression of sHSPs. Although increased *isw-1* expression during developmental stages is adequate to prolong lifespan (Fig. 5a, c), *isw-1* expression has to be robustly reduced from the early development and throughout the adulthood in order to decrease lifespan (Fig. 4a–c). Interestingly, inactivation of yeast *isw2* (the ortholog of *C. elegans isw-1*) prolongs replicative lifespan, which is due to the elevated stress responses[34]. The differential outcome of *isw2* depletion on lifespan compared to *isw-1* knockdown in *C. elegans* could be due to the fact that yeast has two ISWI chromatin-remodeling ATPases, ISW1 and ISW2, which may have compensatory effects regarding longevity. In addition, yeast replicative lifespan is thought to reflect ageing in mitotically active cells, whereas lifespan experiments in *C. elegans* can be considered to monitor the maintenance of non-dividing cells.

Related to life-long *isw-1* RNAi regimen required to shorten lifespan (Figs 3a and 4a–c), mitochondrial stress and associated epigenetic modifications have to occur during development in order to extend lifespan[2,3,35]. One explanation for the role of ISW-1 in lifespan regulation is that it produces and maintains lifespan-promoting epigenetic modifications during both development and adulthood. In this hypothesis, the loss of ISW-1

during development could be compensated by its function during adulthood and vice versa. Similarly, as we see that *isw-1* knockdown within one generation reduces the mitochondria-derived longevity without affecting the wild-type lifespan (Fig. 6b), postdevelopmental *isw-1* RNAi was reported to shorten the lifespan of long-lived insulin/insulin-like growth factor-1 signaling (IIS) mutants[36]. Notably, both mitochondrial stress and reduced IIS result in increased expression of sHSPs (Fig. 1c and ref.[37]), suggesting that ISW-1 is required to promote cytosolic proteostasis during the later stages of life, thereby contributing to the extended longevity in the above-mentioned conditions. sHSPs have been shown to be important in preventing protein aggregation in disease models and promoting lifespan, especially in IIS mutants[31,38,39]. Our data add to these reports by showing that, in addition to the convergence between the transcription factors DAF-16 and HSF-1 on sHSPs, also ISW-1-mediated regulation of sHSP expression is important in longevity.

Mitochondrial dysfunction is buffered by pleiotropic stress response pathways[40]. One of these pathways is the UPR[mt], which promotes mitochondrial proteostasis and lifespan extension by inducing the expression of mitochondrial chaperones and proteases[12,15,28,40]. Our findings demonstrate that, in parallel to the UPR[mt], mitochondrial stress activates a robust cytosolic sHSP-mediated stress response. Previously, it was shown that depletion of the mitochondrial Hsp70 family chaperone, HSP-6, enhances

cytosolic proteostasis via changes in lipid biosynthesis[41]. As *cco-1* RNAi, which was used to stress mitochondria also in our study, does not affect the expression of genes involved in lipid synthesis[41], it is likely that RNAi of *hsp-6* and *cco-1* activate the cytosolic chaperone-mediated response through alternative pathways. Intriguingly, both mitochondrial stress and histone depletion have a similar effect on the expression of histones and chromatin remodelers (Figs 1a, b and 2a–c). As both stressors trigger a common sHSP response, we propose that, in addition to the changes in lipid biosynthesis[41], stressed mitochondria modulate histone levels, which in turn activates ISW-1-mediated chromatin remodeling and HSF-1, resulting in increased expression of sHSPs and prolonged lifespan (Fig. 6f). Importantly, as both mitochondrial and histone stress induces the expression of many chromatin remodelers (Fig. 2a–c), one has to keep in mind that ISW-1 may not be the only chromatin remodeling factor that is required for lifespan extension deriving mitochondrial and histone perturbations.

Why conditions such as mitochondrial or nucleosomal perturbations modulate histone expression, and thereby most likely nucleosome and chromatin structure, is an outstanding question. One plausible explanation follows the model proposed earlier, in which a free soluble pool of histones promotes yeast lifespan by enhancing the removal of post-translationally modified and damaged histones and facilitating the repackaging of nucleosome-free genomic regions into chromatin, thus resulting in more tightly packed chromatin and reduced genomic instability and transcriptional noise[7]. Hence, different stressors could change histone expression to rearrange the epigenetic landscape in order to modulate transcriptional programs. In our model, changes in histone levels increase *isw-1* expression, which then promotes protein folding and disaggregation in the cytosol to protect from concurrent transcriptional noise caused by altered chromatin structure (Fig. 6f). In support of this hypothesis, mitochondrial variability has been shown to produce transcriptional noise[42–44], whereas age-associated loss of histones induces a global increase in transcription[8]. The cytosolic sHSPs are ideally positioned to buffer challenged cytosolic proteostasis and altered cellular function, as they are reported to have widespread effects on stress responses and cell survival, including prevention of protein aggregation, stimulation of protein degradation through the proteasome and autophagy pathways, activation of innate immune response, and inhibition of mitochondrial apoptotic pathways[19].

Although proteostatic defense is based on compartment-specific stress responses, interplay between different cellular proteostasis subnetworks emerged as an essential factor for organismal survival. Mitochondria are at the crossroad between cellular compartments. The work presented here demonstrates that the mitochondria and nucleus share a common stress response, which is associated with changes in histone expression resulting in chromatin remodeling and enhanced cytosolic proteostasis (Fig. 6f). These results underline the importance of chromatin structure and remodeling in orchestrating cellular stress responses to promote longevity.

## Methods

**C. elegans strains and maintenance.** For all experiments, worms were maintained at 20 °C. The N2 (Bristol) strain was used as the wild type. N2, *isw-1(n3297)* (MT16012), *hsf-1(sy441)* (PS3551), HSF-1::GFP (OG497), and HSF-1::HA (OG646) strains were obtained from the Caenorhabditis Genetics Center (CGC). Transcriptional AUW5: *epfEx4[isw-1p::gfp]* and AUW8: *epfIs7[isw-1p::gfp]* reporter strains and AUW7: *epfIs6[isw-1p::isw-1 + myo-2p::gfp]* overexpression strain were generated within this study.

**Plasmids and generation of transgenic C. elegans strains.** *isw-1p::gfp* expression vector was created by amplifying 1563 bp sequence upstream from the transcription start site of *isw-1*-coding region by using *C. elegans* genomic DNA (primers used for cloning are listed in Supplementary Table 3). The PCR product was digested with PciI and AgeI and ligated into the pPD30.38 expression vector containing *gfp* coding sequence cloned between inserted AgeI and NotI restriction sites. Two independent lines carrying *isw-1p::gfp* transgene as extrachromosomal array and one line carrying integrated *isw-1p::gfp* transgene were analyzed in the study. To create vector expressing *isw-1* under its endogenous promoter, *isw-1* ORF was amplified from *C. elegans* cDNA. The PCR product was digested with AgeI and NheI and ligated into the pPD30.38 expression vector containing *isw-1p::gfp* to replace *gfp*-coding sequence between AgeI and NheI restriction sites. Injection marker *myo-2p::gfp* was cloned by amplifying 1179 bp sequence upstream from the transcription start site of *myo-2*-coding region by using *C. elegans* genomic DNA. PCR product was digested with PciI and AgeI and ligated into the pPD30.38 expression vector containing *isw-1p::gfp* to replace *isw-1* promoter sequence between PciI and AgeI restriction sites. All transgenic strains were created by using microinjection. *isw-1p::gfp* and *isw-1p::isw-1* extrachromosomal arrays were integrated into the genome by using gamma irradiation. After integration, *isw-1* transcriptional strain and *isw-1* overexpression strain were backcrossed three and six times with the wild type, respectively.

**RNAi and doxycycline treatment.** Bacterial clones for RNAi (*cco-1*, F26E4.9; *his-3*, T10C6.12; *mrps-5*, E02A10.1; *swsn-3*, Y71H2AM.17; *isw-1*, F37A4.8; *nurf-1*, F26H11.2; *mep-1*, M04B2.1; *dve-1*, ZK1193.5; *ubl-5*, F46F11.4; *hsp-4*, F43E2.8) were taken from the Ahringer RNAi library (https://www.sourcebioscience.com/products/life-science-research/clones/rnai-resources/c-elegans-rnai-collection-ahringer/), except *atfs-1* (ZC376.7) RNAi clone, which was taken from Vidal RNAi library (https://www.sourcebioscience.com/products/life-science-research/clones/rnai-resources/c-elegans-orf-rnai-resource-vidal/). In chromatin remodeler RNAi lifespan experiments with RNAi regimen i, L2–L3 worms of P0 generation were put to RNAi plates, and progeny from these worms (F1 generation) were used for lifespan. In other lifespan and fluorescent worm imaging experiments, gravid adult worms (P0 generation) were let to lay eggs on treatment plates, and this generation (F1) was used for experiments. In qRT-PCR and proteasome activity assays, worms were synchronized by bleaching, let to hatch overnight in M9 solution, and plated to the indicated RNAi plates as L1 larvae. In experiments that used day 2 adult worms (*isw-1* expression analysis of day 2 adult worms and proteasome activity assays), worms were transferred to plates containing 10 μM of 5-Fluorouracil (Sigma) at the L4 larval stage in order to prevent progeny production. In experiments with doxycycline, worms were grown on plates containing doxycycline (15 μg/ml) (Sigma) from L1 larvae.

**Lifespan analysis.** *C. elegans* lifespan experiments were carried out at 20 °C. To prevent progeny production, worms were transferred to plates containing 10 μM of 5-Fluorouracil (Sigma) at the L4 larval stage. Worms that had exploded vulva or that crawled off the plate were censored. Worms were counted as dead if there was no movement after poking with a platinum wire. Lifespans were checked every 1–2 days. For lifespan data, mean lifespan ± standard error (s.e.) is reported (Supplementary Table 2). Statistical significances were calculated by using the log-rank (Mantel-Cox) method.

**RNA sequencing.** Total RNA was extracted by using TriPure Isolation Reagent (Roche) and assessed for degradation using an Advanced Analytical Agilent Fragment Analyzer. All 9 samples have an RNA Quality Number >9.6. Ilumina Truseq stranded polyA-mRNA library was prepared and sequenced for 100 cycles at the Genome Technologies Facility at the University of Lausanne. The 9 samples were multiplexed and sequenced on one lane of an Illumina Hiseq 2500, yielding a minimum of 26 million reads per sample. All obtained results passed the FastQC quality check and were subsequently mapped to the *C. elegans* reference genome using STAR aligner (version 2.5.3a). The Ensembl Release 83[45] genome fasta and GTF annotation files were used as the reference. There was an average of ~90% uniquely mapped reads. Counts per gene were obtained using HTseq-count[46] (version 0.7.2). Genes with >5 reads in >3 samples were retained in the analysis. The limma package[47] was used for differential expression, where two contrasts were assigned: the *cco-1* RNAi effect (*cco-1* RNAi vs. EV) and the *his-3* RNAi effect (*his-3* RNAi vs. EV). Genes with a Benjamini–Hochberg adjusted *P*-value <0.05 and an absolute log fold change of 2 were considered differentially expressed. GSEA was performed using the clusterProfiler package[48]. GO genesets in gmt format were obtained from GO2MSIG (release of April 2015). We added some custom genesets to the biological process terms list (Supplementary Table 1). For each treatment, all expressed genes were ordered by decreasing fold change based on the differential expression analysis. For each analysis and GO category, we performed 100,000 permutations using the "fgsea" option, a minimum gene set size of 5, and a maximum of 1000. False discovery rate-adjusted *P*-values were calculated using the Benjamini–Hochberg method.

**Quantitative RT-PCR.** Worms were synchronized by bleaching and plated to RNAi plates as L1 larvae. Worms were collected from plates with M9 solution and freezed immediately in liquid nitrogen minimizing the time worms stayed in the solution. NucleoSpin RNA Isolation Kit (Macherey-Nagel) was used for RNA

extraction from embryos and L1 larvae. TriPure Isolation Reagent (Roche) was used to extract RNA from L4 larvae and day 1 and day 2 adult worms. cDNA synthesis was carried out with the QuantiTect Reverse Transcription Kit (Qiagen) and qRT-PCR reactions were run with the SYBR Green reagent (Roche) using Lightcycler 480 (Roche). qRT-PCR data were normalized to the expression of act-1 and pmp-3. qRT-PCR oligos used in this study are provided in Supplementary Table 4. Statistical significances were analyzed by using Student's t-test or one-way analysis of variance (ANOVA).

**C. elegans imaging**. GFP expression profile of isw-1p::gfp strain was imaged in L4 larvae by using Zeiss Axioplan microscope using 10 × 0.3 NA objective. his-3, his-8, cco-1, mrps-5, atfs-1, dve-1, ubl-5, and hsp-4 RNAi and doxycycline-, heat-shock-, and NaCl-treated isw-1p::gfp strain was imaged at day 1 of adulthood with a standard stereomicroscope with fluorescent light source (Nikon). For all imaging, worms were paralyzed with 10 mM tetramisole (Sigma).

**Heat stress and osmotic stress**. In western blot and isw-1p::gfp transcriptional reporter assays, worm plates were put to 37 °C for 90 min. Worms were lysed/imaged after 4 h recovery period in 20 °C. For western blot, heat shock was given at L4 larval stage and, for isw-1p::gfp reporter worms, at day 1 of adulthood. For osmotic stress, worms were cultured from hatching on plates containing 250 mM NaCl.

**In vitro proteasome activity assay**. Day 2 adult worms were used for proteasome activity assays. For these experiments, freshly collected worms were lysed in proteasome activity assay buffer (50 mM Tris-HCl, pH 7.4, 250 mM sucrose, 5 mM MgCl2, 0.5 mM EDTA, 0.5 mM ATP, and 1 mM dithiothreitol) using plastic pestle fit for 1.5 ml Eppendorf tube. Lysate was centrifuged at 15,000×g for 15 min at 4 °C. In all, 10 μg of total protein was used in assays with 100 μM of Suc-Leu-Leu-Val-Tyr-AMC (Bachem) to measure chymotrypsin-like proteasome activity. Proteasome inhibitor MG132 was used with 25 μM concentration. Peptide cleavage was measured with VICTOR × 4 plate reader (PerkinElmer) using black OptiPlate-96 F-plates (PerkinElmer). Statistical significances were analyzed by using one-way ANOVA.

**Immunoblot analysis**. Worms were synchronized by bleaching and plated to RNAi plates as L1 larvae. Worms were collected from plates at L4 stage with M9 solution and freezed immediately in liquid nitrogen minimizing the time worms stayed in the solution. Worms were lysed in RIPA buffer (50 mM Tris-HCl (pH 7.4), 150 mM NaCl, 1 mM EDTA, 1% NP-40, 1% sodium deoxycholate, 0,1% sodium dodecyl sulfate (SDS)) with sonication. Lysates from L4 larvae worms were resolved by SDS–polyacrylamide gel electrophoresis. Histone H2A (ab88770, used with 1:400 dilution), H2B (ab52484, used with 1:1000 dilution), and H3 (ab1791, used with 1:1000 dilution) antibodies were purchased from Abcam, histone H4 antibody (sc-10810, used with 1:400 dilution) from Santa Cruz, and α-tubulin antibody (T5168, used with 1:1000 dilution) from Sigma. Uncropped membrane slices are shown in Supplementary Fig. 6.

**Chromatin immunoprecipitation**. Worms were synchronized by bleaching and plated to RNAi plates as L1 larvae. For each RNAi condition, 15,000 worms (three biological replicates, 5000 per replicate) were used. Worms were collected from plates at L4 stage with M9 solution and crosslinked for 30 min with 2% formaldehyde under rocking. After crosslinking, worms were incubated in 0.125 M glycine for 15 min under rocking. Worms were sonicated in RIPA buffer: samples from the experiment with his-3 RNAi and HSF-1::HA were sonicated with probe sonicator, and samples from the experiment with cco-1 RNAi and HSF-1::GFP were sonicated with Covaris sonicator. Both methods produced DNA fragments of approximately 100–500 bp. HSF-1::HA was immunoprecipitated with anti-HA-agarose (A2095, Sigma, 50 μl used for each pulldown) and HSF-1::GFP with GFP antibody (Abcam, ab290, 3 μg used for each pulldown) and Protein G Dynabeads (ThermoFisher). DNA fragments were extracted with phenol–chloroform–isoamylalcohol and precipitated with ethanol. qRT-PCR primers used to quantify precipitated DNA fragments are listed in Supplementary Table 4.

**Quantification and statistical analysis**. Statistical tests and the sample sizes were chosen based on studies with similar experimental design. Sample sizes were chosen without performing statistical tests. Data meet the assumption for each statistical test used. Randomized or blinded experiments were not performed. For all bar graphs, mean ± standard deviation (s.d.) is represented.

**Data availability**. Data supporting the results reported in this article are available from the corresponding author upon request. Gene expression data have been deposited in GEO under accession code GSE105036.

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

## Acknowledgements

O.M is supported by FEBS long-term fellowship. P.M.Q. was supported by a long-term fellowship from EMBO (ALTF 480-2014). This work is supported by grants from the École Polytechnique Fédérale de Lausanne, the Swiss National Science Foundation (31003A-140780), the AgingX program of the Swiss Initiative for Systems Biology (51RTP0-151019), and the NIH (R01AG043930). The authors thank Carina Holmberg and Pierre Gönczy for sharing reagents and Laurent Mouchiroud and Vincenzo Sorrentino for assistance with the experiments. Some strains were provided by the CGC, which is funded by NIH Office of Research Infrastructure Programs (P40 OD010440).

## Author contributions

O.M. and J.A. designed the project. O.M. performed the experiments and analyzed the data. M.S.B.S analyzed the RNA-seq data. P.M.Q analyzed the *C. elegans* proteomic data set. S.M.D.A.G. supervised the experiments. O.M. and J.A. wrote the manuscript. All authors commented on the manuscript.

## Additional information

**Competing interests:** The authors declare no competing financial interests.

