## [Peer Review File · Nature Communications]

Reviewers' Comments:

Reviewer #1:

Remarks to the Author:

This manuscript reports the interesting findings that mitochondrial stress results in altered histone levels and an upregulation of the histone remodeler ISW-1. The authors further showed that increased dosage of ISW-1 can increase lifespan and that depends on the increased expression of several small heat shock proteins. The authors concluded that mitochondrial dysfunction can engage the cytosolic stress pathways via chromatin changes.

Specific comments:

- 1) Does his-3 RNAi affects lifespan and is that isw-1 dependent?
- 2) ISW-1 seems to respond to a wide variety of stressors. Have the authors found a stress that does NOT induce ISW-1 (e.g. heat stress, starvation, osmotic stress)? i.e. how specific is the ISW-1 response?
- 3) Similarly, how specific is the response of altered histone gene expression upon stress? Does that happen under many different stress conditions, OR specifically under mitochondrial stress?
- 4) While RNAi of the various sHSPs was sufficient to block the lifespan increase of the ISW-1 OE worms, does RNAi of the various sHSP suppress the lifespan of the cco-1 RNAi or other mitochondrial dysfunction worms as well?
- 5) What happen in worms where both his-3 and cco-1 are RNAi knockdown?

Minor comments:

- Line 62, the authors indicated that in yeast, altered histone levels lead to mitochondrial dysfunctions. Does that happen in *C. elegans* as well? e.g. Does his-3 RNAi lead to changes in mitochondrial function / stress response?
- Line 130, the authors stated that the upregulation of ISW-1 suggests a compensatory response. This seems too speculative in the absence of further data.

Reviewer #2:

Remarks to the Author:

Review of the manuscript "The chromatin remodelling factor ISW-1 integrates responses against nuclear and mitochondrial stress", by Matilainen et al. for Nature Communications

In the manuscript presented by Matilainen and colleagues, the authors show by qRT-PCR experiments in *C. elegans* that mitochondrial perturbations lead to altered expression of various histones and an induction of several chromatin remodellers, including ISW-1. Similarly, when knocking down histone H2A, they observe a compensatory upregulation of histones and remodelling enzymes. Along with these changes, cytosolic small heat shock proteins (sHSPs) are being upregulated, an upregulation that largely depends on the chromatin remodeller ISW-1 and the transcription factor HSF-1. The authors eventually show that the longevity caused by mitochondrial perturbations depends on ISW-1 and on the upregulation of sHSPs, and that ISW-1 is sufficient to drive this process.

Overall, the manuscript supports the idea that a disturbed histone balance, which for example can be induced by mitochondrial perturbations, triggers compensatory histone expression and induction of the chromatin remodeller ISW-1, of which the latter induces a cytoplasmic proteostatic stress response, which enhances stress resistance and extends the lifespan of the organism.

As the authors already state in the manuscript, the link between chromatin remodelling, stress response, and lifespan is not completely new. Other recent studies have already shown that different chromatin remodellers take part in the response to environmental stress and are able to regulate lifespan in different organisms such as yeast and *C. elegans*. Most importantly,

inactivation of *isw2* in yeast, the ortholog of *C. elegans isw-1*, extends replicative lifespan, and the chromatin remodeller *swi/snf* has been found required for longevity of insulin signalling mutants in *C. elegans*. Also, the fact that mitochondrial perturbations can not only trigger UPR_{mito} but also induce cytosolic heat shock response by expression of sHSPs has been shown before (Kim et al., Cell 2016). And it had already been observed that mitochondrial perturbations can influence the nuclear epigenome, at the example of DNA methylation (reviewed in D'Aquila et al., Biogerontology 2015).

The actual novelty of the manuscript lies now in three observations: 1) That mitochondrial perturbations may lead to not only changes in the nuclear epigenome but an actual disturbance of nuclear chromatin, i.e. changes in histone and chromatin remodeller expression, one of the remodellers being ISW-1. 2) That induction of the cytosolic heat shock response due to mitochondrial perturbations is dependent on ISW-1. 3) That ISW-1 OE is sufficient to trigger this cytosolic heat shock response.

Overall, this manuscript provides an interesting new example of how chromatin changes and particularly chromatin remodeller activities are important for stress responsive and lifespan regulatory gene expression events, which makes this manuscript appealing for the general aging research community. However, the mechanisms proposed in this manuscript seem insufficiently supported, and thus I would recommend publication in a more specialized journal, unless more depth is provided and the Major Comments below have been addressed.

Major Comments:

1. One of the main conclusions of the manuscript is that histone and mitochondrial perturbations cause a similar stress response in the organisms, via the actions of ISW-1 and HSF-1 and the activation of sHSPs. The data supporting this conclusion is mainly derived from qRT-PCR experiments on selected genes, and, although it seems that both perturbations (*cco-1* and *his-3* RNAi) affect expression of those genes in a similar way (Fig. 1, Fig. S1), some differences are still observed. To more reliably conclude that both perturbations cause a similar response and to gain a better understanding of the pathways by which mitochondrial and histone perturbations affect stress responses, I suggest to investigate gene expression across the genome, e.g. by mRNA-seq experiments, in both conditions (*cco-1* RNAi and *his-3* RNAi). And conducting these experiments also in *isw-1* and *hsf-1* mutants would help to decipher the involvement of ISW-1 and HSF-1 in these responses.
 2. The authors claim that *cco-1* RNAi extends lifespan by regulation of histones, the upregulation of ISW-1, and the resulting induction of sHSPs. They also propose that *his-3* RNAi triggers the same downstream effects. If this is true, one would assume that also *his-3* RNAi should extend lifespan. The authors should test this.
 3. In Fig. S1, changes in the expression of many genes, although statistically significant by Student's t-test, still appear to be mild. The authors should clarify, if these trends are reproducible between different independent experiments.
 4. The resulting histone composition after *cco-1* RNAi and *his-3* RNAi shows some differences, according to the mRNA expression data. Only in *his-3* RNAi all core histones are overexpressed, while overexpression may be limited to H3 for *cco-1* RNAi. It would be desirable to quantify all histones (not only H3) at the protein level under *cco-1* RNAi and *his-3* RNAi, to obtain a clearer view of the overall effects and any possible differences caused by the two perturbations with regard to the histone/chromatin composition.
 5. The authors show that H1 linker histone is by far the most upregulated histone under the two perturbations tested. The authors should test, if the downstream phenotypes depend on H1 upregulation.
- H1 has a central role in chromatin organization and higher-order chromatin structure. Although this would go beyond the scope of this revision, it would be really interesting to determine how chromatin environment and accessibility change in response to *cco-1* RNAi and *his-3* RNAi. For

example, MNase digestion based techniques could help in determining this.

See Corona et al., PLoS Biol 2007.

6. It would be desirable to comprehensively test which genes are (directly) regulated by ISW-1 in response to stress conditions. Does ISW-1 indeed help HSF-1 to regulate its target genes? A combination of mRNA-seq experiments and ChIP-seq experiments for ISW-1 and HSF-1 in response to the different perturbations should help a lot to clarify the mechanism by which ISW-1 regulates the cytosolic stress response.

7. The authors suggest an order of events, in which mitochondrial stress causes changes in histone expression, which leads to an increase in ISW-1 expression, which eventually triggers sHSP expression and extends lifespan. To properly support this order of events, histone expression under *cco-1* RNAi should be checked in *isw-1* mutant conditions. If the model is correct, then *isw-1* mutation should not affect histone expression.

8. As a model of mitochondrial ETC perturbation, the authors exclusively use *cco-1* RNAi. To make sure that the findings are not *cco-1* RNAi specific, the authors should validate their finding of an *isw-1*-dependent longevity in a different mitochondrial perturbation model, e.g. in *isp-1* mutants.

Minor Comments:

9. Given that also other chromatin remodelers are upregulated in response to histone and mitochondrial stress, and some of them may affect lifespan, the authors should discuss which is the contribution of those other remodelers beside ISW-1 in the response to *cco-1* RNAi and *his-3* RNAi, as they did not go further in studying them.

10. There are several other long-lived *C. elegans* models, whose longevity depends on the actions of sHSPs. The authors should at least discuss, if they think that *isw-1* is involved in these other models.

11. The third section of the "results" section is very long compared with others. I would recommend to split it into two parts for better readability.

Reviewer's comments:

We thank you for your effort in reading and commenting on this manuscript. We have performed multiple experiments based on your remarks, and we feel that this has significantly improved our manuscript.

Reviewer #1

Specific comments:

1) Does his-3 RNAi affects lifespan and is that isw-1 dependent?

This is a good point from the reviewer. Lifespan experiments demonstrate that in wild-type worms (N2), *his-3* RNAi slightly, but significantly and reproducibly, extends lifespan. However, *his-3* RNAi does not affect the lifespan of *isw-1(n3297)* and *hsf-1(sy441)* mutant worms, demonstrating that both ISW-1 and HSF-1 are required for the lifespan extension deriving from *his-3* RNAi. These results are now included to the manuscript (Figs 6c-d and Supplementary Fig 5f, Supplementary Table 1). In addition to full *his-3* RNAi, also combined EV/*his-3* RNAi extended lifespan (Fig. 6e, Supplementary Table 1).

2) ISW-1 seems to respond to a wide variety of stressors. Have the authors found a stress that does NOT induce ISW-1 (e.g. heat stress, starvation, osmotic stress)? i.e. how specific is the ISW-1 response?

It is true that ISW-1 responds to many stressors, which may be an indicator for the central role of chromatin remodeling in organismal stress responses. In addition, to histone- and mitochondrial stress, we treated *isw-1p::gfp* reporter worms with *hsp-4* RNAi to induce ER stress, and exposed them to heat stress (90 min at 37 °C) and osmotic stress (NaCl with final concentration of 250 mM on plates). Heat stress did not increase the *isw-1* expression, whereas *hsp-4* RNAi caused a mild increase in fluorescence (Fig. 3e). However, the increase in GFP signal observed upon *hsp-4* RNAi is minor compared to that observed with treatments with *cco-1* or *his-3* RNAi (Fig. 3e). Interestingly, in addition to mitochondrial- and histone stress, we found that also osmotic stress is a strong inducer of *isw-1p::gfp* expression (Fig. 3e). This ISW-1 response to changing osmolality may be due to the altered histone levels (see next point and Fig. 3f).

3) Similarly, how specific is the response of altered histone gene expression upon stress? Does that happen under many different stress conditions, OR specifically under mitochondrial stress?

We investigated the protein levels of H2A, H2B and H3, which show changes upon mitochondrial/histone stress (Fig. 1a), ER stress, heat stress and osmotic stress (Fig. 3f). Although ER stress (*hsp-4* RNAi) and heat stress slightly increase the levels of H2A and H2B, osmotic stress strongly modulates levels of H2A, H2B and H3 histones (Fig. 3f). Accordingly and as mentioned within the previous point, osmotic stress is a strong inducer of *isw-1p::gfp* (Fig. 3e). Altogether, our data indicate that *isw-1* expression is induced upon stressors that modulate histone levels.

4) While RNAi of the various sHSPs was sufficient to block the lifespan increase of the ISW-1 OE worms, does RNAi of the various sHSP suppress the lifespan of the *cco-1* RNAi or other mitochondrial dysfunction worms as well?

We examined the role of sHSPs in *cco-1* RNAi-mediated longevity, as this stressor is widely used in this manuscript. We performed two lifespans with combined RNAi of *cco-1/hsp-16.1*, *hsp-16.2*, *hsp-16.41*, and *hsp-16.48*. In these experiments, the volume of *cco-1* RNAi bacteria in combined RNAi was 50 % and each *hsp* RNAi bacteria 12,5 %. The combined *cco-1/4hsp* RNAi did not reduce the lifespan extension deriving from EV/*cco-1* RNAi (Reply to Reviewers Fig. 1). This suggests that other (e.g. mitochondrial) chaperones play a predominant role in the longevity upon *cco-1* RNAi feeding¹. Alternatively, it is possible that this combined RNAi did not sufficiently reduce sHSP expression upon *cco-1* RNAi. However, as the combined sHSP RNAi (each 25 % of the total RNAi volume) reduced the lifespan of *isw-1* OE strain, we are certain that sHSPs are important factors determining ISW-1-mediated longevity (see Fig. 5d).

Reply to Reviewers Fig. 1. Lifespan of *cco-1/hsp-16.1*, *hsp-16.2*, *hsp-16.41*, *hsp-16.48* RNAi treated worms compared to that of EV/*cco-1* RNAi treated worms.

5) What happen in worms where both *his-3* and *cco-1* are RNAi knockdown?

Although EV/*his-3* RNAi extended lifespan, combined *cco-1/his-3* RNAi did not further extend the longevity of EV/*cco-1* RNAi treated worms (Fig. 6e, Supplementary Table 1). This indicates that *his-3* and *cco-1* RNAi affect longevity through similar mechanisms.

Minor comments:

Line 62, the authors indicated that in yeast, altered histone levels lead to mitochondrial dysfunctions. Does that happen in *C. elegans* as well? e.g. Does *his-3* RNAi lead to changes in mitochondrial function / stress response?

Galdieri *et al.* demonstrated that decreased histone expression and deficient nucleosome assembly increases mitochondrial DNA copy number, oxygen consumption, ATP synthesis and oxidative phosphorylation in yeast (Galdieri *et al.*, *Mol Cell Biol.*, 2016)². Our qPCR data showed that *his-3* RNAi does not trigger the mitochondrial unfolded protein response (UPR^{mt}) (Fig. 1d). However, when detecting OXPHOS subunits with Western blot, we found that *his-3* RNAi slightly elevates the protein levels of some OXPHOS subunits

(Reply to Reviewers Fig. 2). As our manuscript focuses on cytosolic stress response instead of mitochondrial function and as more in depth analysis would be required to elucidate the effect of histone stress on mitochondrial function in *C. elegans* (e.g. role of ISW-1 and HSF-1 on the regulation of the mitochondrial network and biogenesis/mitophagy), we feel that this data diverges from the central message of the paper, and therefore should not be included to the manuscript.

Reply to Reviewers Fig. 2. OXPHOS subunits blotted with with Abcam OXPHOS antibody cocktail (ab110413). We were unable to detect complex four (CIV) subunits with this antibody cocktail.

Line 130, the authors stated that the upregulation of ISW-1 suggests a compensatory response. This seems too speculative in the absence of further data.

We agree with the reviewer that our statement about the compensation is too speculative and removed it in the revised manuscript.

Reviewer #2

Review of the manuscript “The chromatin remodelling factor ISW-1 integrates responses against nuclear and mitochondrial stress”, by Matilainen et al. for Nature Communications

In the manuscript presented by Matilainen and colleagues, the authors show by qRT-PCR experiments in *C. elegans* that mitochondrial perturbations lead to altered expression of various histones and an induction of several chromatin remodellers, including ISW-1. Similarly, when knocking down histone H2A, they observe a compensatory upregulation of histones and remodelling enzymes. Along with these changes, cytosolic small heat shock proteins (sHSPs) are being upregulated, an upregulation that largely depends on the chromatin remodeller ISW-1 and the transcription factor HSF-1. The authors eventually show that the longevity caused by mitochondrial perturbations depends on ISW-1 and on the upregulation of sHSPs, and that ISW-1 is sufficient to drive this process.

Overall, the manuscript supports the idea that a disturbed histone balance, which for example can be induced by mitochondrial perturbations, triggers compensatory histone expression and induction of the chromatin remodeller ISW-1, of which the latter induces a cytoplasmic proteostatic stress response, which enhances stress resistance and extends the lifespan of the organism.

As the authors already state in the manuscript, the link between chromatin remodelling, stress response, and lifespan is not completely new. Other recent studies have already shown that different chromatin remodellers take part in the response to environmental

stress and are able to regulate lifespan in different organisms such as yeast and *C. elegans*. Most importantly, inactivation of *isw2* in yeast, the ortholog of *C. elegans isw-1*, extends replicative lifespan, and the chromatin remodeller *swi/snf* has been found required for longevity of insulin signalling mutants in *C. elegans*. Also, the fact that mitochondrial perturbations can not only trigger UPR_{mito} but also induce cytosolic heat shock response by expression of sHSPs has been shown before (Kim *et al.*, *Cell* 2016). And it had already been observed that mitochondrial perturbations can influence the nuclear epigenome, at the example of DNA methylation (reviewed in D'Aquila *et al.*, *Biogerontology* 2015).

The actual novelty of the manuscript lies now in three observations: 1) That mitochondrial perturbations may lead to not only changes in the nuclear epigenome but an actual disturbance of nuclear chromatin, i.e. changes in histone and chromatin remodeller expression, one of the remodellers being ISW-1. 2) That induction of the cytosolic heat shock response due to mitochondrial perturbations is dependent on ISW-1. 3) That ISW-1 OE is sufficient to trigger this cytosolic heat shock response.

Overall, this manuscript provides an interesting new example of how chromatin changes and particularly chromatin remodeller activities are important for stress responsive and lifespan regulatory gene expression events, which makes this manuscript appealing for the general aging research community. However, the mechanisms proposed in this manuscript seem insufficiently supported, and thus I would recommend publication in a more specialized journal, unless more depth is provided and the Major Comments below have been addressed.

We appreciate the reviewer's view to place our manuscript within the context of the current literature. However, we take this opportunity to comment some of the statements regarding the novelty of this manuscript.

As we discuss in the manuscript (line 322), inactivation of *isw2* in yeast affects longevity in an opposite manner compared to *C. elegans* (Dang *et al.*, *Cell Metab.*, 2014)³. We think that our finding demonstrating the requirement of *isw-1* for normal lifespan in a multicellular organism is important, and at the same time highlights the differences between yeast and *C. elegans* in lifespan studies.

Riedel *et al.*, *Nat Cell Biol.*, 2013 showed that the chromatin remodeler SWI/SNF regulates lifespan especially in long-lived insulin signaling mutants, and they elegantly demonstrate that SWI/SNF functions together with the transcription factor DAF-16. Instead of insulin signaling, our manuscript focuses on linking histone- and mitochondrial stress with chromatin remodeling factors, chaperone expression and longevity. Despite these differences, the Riedel *et al.* study and our work converge and support the importance of chromatin remodelers in stress responses and longevity.

Kim *et al.*, *Cell*, 2016⁴ demonstrated that mitochondrial perturbation via *hsp-6* RNAi results in changes in lipid signaling, which then triggers a cytosolic stress response. Therefore, we agree that mitochondria-to-cytosol stress response described in our manuscript is not conceptually novel. As discussed in our manuscript (line 349), Kim *et al.*, however, showed that *cco-1* RNAi does not induce genes involved in lipid biosynthesis. Thus, the proposed mechanism of how mitochondrial stress is translated into cytosolic response described in our study is novel. Again, in our opinion work by Kim *et al.* and our studies complement

each other by providing distinct mechanisms of how mitochondrial perturbations result in enhancement of cytosolic protein homeostasis.

We furthermore agree with the reviewer that the relationship between mitochondrial perturbations and the nuclear epigenome has been studied previously. However, to our knowledge it has not been previously shown that mitochondrial stress modulates histone levels. Therefore, we are convinced that our study provides a novel link between mitochondria and the epigenome.

Major Comments:

1. One of the main conclusions of the manuscript is that histone and mitochondrial perturbations cause a similar stress response in the organisms, via the actions of ISW-1 and HSF-1 and the activation of sHSPs. The data supporting this conclusion is mainly derived from qRT-PCR experiments on selected genes, and, although it seems that both perturbations (*cco-1* and *his-3* RNAi) affect expression of those genes in a similar way (Fig. 1, Fig. S1), some differences are still observed. To more reliably conclude that both perturbations cause a similar response and to gain a better understanding of the pathways by which mitochondrial and histone perturbations affect stress responses, I suggest to investigate gene expression across the genome, e.g. by mRNA-seq experiments, in both conditions (*cco-1* RNAi and *his-3* RNAi). And conducting these experiments also in *isw-1* and *hsf-1* mutants would help to decipher the involvement of ISW-1 and HSF-1 in these responses.

We appreciate the reviewer's view on how to strengthen the connection between histone- and mitochondrial perturbations. In the manuscript we did not claim that global responses deriving from *cco-1* and *his-3* RNAi are similar, we concentrated only on the expression of sHSPs. To examine globally which responses are common between histone- and mitochondrial perturbations, we performed RNA-sequencing with *cco-1* and *his-3* RNAi treated worms. When examining pathways affected by *cco-1* and *his-3* RNAi by using GO enrichment analysis, we found that these stressors each modulate many biological processes that are not common (Supplemental Fig. 1c). This is not surprising, as these RNAi treatments target different cellular compartments. However, we found that both *cco-1* and *his-3* RNAi enrich the innate immune response GO category (Supplemental Fig. 1c). Notably, mitochondrial stress is known to enhance innate immunity in *C. elegans* (Pellegrino *et al.*, *Nature*, 2014)⁵, and our RNA-seq validates these results. Interestingly, *his-3* RNAi leads to even stronger enrichment of innate immune response (Supplemental Fig. 1c), provoking the question whether changes in histone balance are activating innate immunity also upon mitochondrial stress. The common induction of the innate immune response is interesting, but given the focus of our work on the sHSP response, we feel that addressing this would be beyond the scope of this paper.

2. The authors claim that *cco-1* RNAi extends lifespan by regulation of histones, the upregulation of ISW-1, and the resulting induction of sHSPs. They also propose that *his-3* RNAi triggers the same downstream effects. If this is true, one would assume that also *his-3* RNAi should extend lifespan. The authors should test this.

This is a good point, and comment also from another reviewer. Lifespan experiments demonstrate that in wild-type worms (N2), *his-3* RNAi slightly, but significantly and reproducibly, extends lifespan. However, *his-3* RNAi does not affect the lifespan of *isw-*

1(*n3297*) and *hsf-1*(*sy441*) mutant worms, demonstrating that both ISW-1 and HSF-1 are required for the lifespan extension deriving from *his-3* RNAi. Additionally, also combined EV/*his-3* RNAi extends lifespan. These results are now included in the revised manuscript (Figs 6c-e and Supplementary Fig. 5f, Supplementary Table 1).

3. In Fig. S1, changes in the expression of many genes, although statistically significant by Student's t-test, still appear to be mild. The authors should clarify, if these trends are reproducible between different independent experiments.

These trends are reproducible between different experiments, as demonstrated below (Reply to Reviewers Fig. 3), where we analyzed the expression of chromatin remodelers in a separate independent experiment. There are minor differences between these data and the results shown in Fig. 2b-c, but the main trend, which is that both histone- and mitochondrial stress induce the expression of multiple chromatin remodeling enzymes, remains the same. The global increase in the expression of chromatin remodelers can furthermore also be seen from the new RNA-seq data (Fig. 2a).

Reply to Reviewers Fig. 3. Independent repeats from experiments presented in Fig. 2b-c.

4. The resulting histone composition after *cco-1* RNAi and *his-3* RNAi shows some differences, according to the mRNA expression data. Only in *his-3* RNAi all core histones are overexpressed, while overexpression may be limited to H3 for *cco-1* RNAi. It would be desirable to quantify all histones (not only H3) at the protein level under *cco-1* RNAi and *his-3* RNAi, to obtain a clearer view of the overall effects and any possible differences caused by the two perturbations with regard to the histone/chromatin composition.

According to the reviewer's comment, we now also analyzed histone H2A, H2B and H4 expression by Western blots (in addition to H3 levels) and added this information in the revised Fig. 1a. *cco-1* RNAi does not seem to affect histone H2A protein level, whereas *his-3* RNAi seems to slightly decrease it. *his-3* RNAi targets 9 different histone H2A genes, which may be the reason for decreased histone H2A protein levels. Protein levels of histone H2B are increased upon both *cco-1* and *his-3* RNAi, whereas histone H4 remains unchanged. Altogether, both treatments are affecting histone levels, although not in a completely similar manner. In addition, histone H3 seems to be the most affected histone according to these Western blot experiments.

5. The authors show that H1 linker histone is by far the most upregulated histone under the two perturbations tested. The authors should test, if the downstream phenotypes depend on H1 upregulation. H1 has a central role in chromatin organization and higher-order chromatin structure. Although this would go beyond the scope of this revision, it would be really interesting to determine how chromatin environment and accessibility change in response to *cco-1* RNAi and *his-3* RNAi. For example, MNase digestion based techniques could help in determining this. See Corona et al., PLoS Biol 2007.

To test if the H1 linker histone HIL-1 is contributing to downstream phenotypes during mitochondrial stress, we treated worms with EV, EV/*cco-1* and *hil-1/cco-1* RNAi, and examined the expression of sHSPs. *hil-1* RNAi did not affect the induction of sHSPs upon mitochondrial stress. However, we found that *hil-1* RNAi did not affect *hil-1* mRNA level (Reply to Reviewers Fig. 4a), although the same RNAi clone from Ahringer library has been used previously (Pothof *et al.*, *Genes Dev.*, 2003⁶). *hil-1* is expressed at relatively low level (qPCR Cp values around 34-36), which adds uncertainty to its expression profiling with qPCR.

As the *isw-1* OE strain shows similar phenotypes as the *cco-1* and *his-3* RNAi treated worms, we tested whether the expression of *hil-1* is also increased in this strain. Unlike within *cco-1* and *his-3* RNAi treated worms, the expression of *hil-1* is not increased upon *isw-1* overexpression (Reply to Reviewers Fig. 4b). Therefore, it is unlikely that the downstream phenotypes observed in this manuscript are dependent on the H1 linker histone *hil-1*. Although not a comment from the reviewers, we examined how increased *isw-1* expression affects histone H3 protein levels. Western blot analysis showed that the *isw-1* OE strain has similar histone H3 expression as N2 worms (Supplementary Fig. 4e), thereby indicating that ISW-1 is not regulating histone levels.

We agree with the reviewer that determining chromatin environment and accessibility with MNase digestion would be an interesting avenue for further study. However, we also agree that this is beyond the scope of this revision, as these experiments would take a substantial amount of time.

Reply to Reviewers Fig. 4. (a) *hil-1* and sHSP expression after treatment of N2 worms with EV, EV/*cco-1* and *hil-1/cco-1* RNAi. Bars represent mRNA level relative empty vector with error bars indicating mean \pm s.d. (* $P < 0.05$; ** $P < 0.01$; *** $P < 0.001$; one-way ANOVA with Dunnett's post test). (b) *hil-1* expression in *isw-1* OE strain. Bars represent mRNA level relative N2 with error bars indicating mean \pm s.d. (** $P < 0.01$; unpaired Student's *t*-test). In (a) and (b) worms were collected for qRT-PCR analysis at L4 stage.

6. It would be desirable to comprehensively test which genes are (directly) regulated by ISW-1 in response to stress conditions. Does ISW-1 indeed help HSF-1 to regulate its target genes? A combination of mRNA-seq experiments and ChIP-seq experiments for ISW-1 and HSF-1 in response to the different perturbations should help a lot to clarify the mechanism by which ISW-1 regulates the cytosolic stress response.

As our manuscript mainly focuses on the role of ISW-1 in the proteostatic sHSP-response and longevity upon histone- and mitochondrial stress, we think that further exploration of ISW-1-regulated genes would confuse and not improve this manuscript. Likewise, in our opinion HSF-1 and ISW-1 ChIP-seq experiments with different stress conditions are not in line with the sHSP-centric focus of our work. Moreover, ISW-1 ChIP would require construction and validation of a strain expressing tagged ISW-1, as there are no antibodies for endogenous *C. elegans* ISW-1.

To test whether ISW-1 assists HSF-1 to regulate its target genes, we performed HSF-1 ChIP-qPCR. Importantly, the four sHSPs (*hsp-16.1*, *hsp-16.2*, *hsp-16.41* and *hsp-16.48*) are well known targets of HSF-1, and therefore we examined whether *isw-1* RNAi affects HSF-1 binding on the sHSP promoters upon histone- or mitochondrial stress. For these experiments we used worms expressing tagged HSF-1, i.e. HSF-1::GFP worms fed with EV, EV/*cco-1* or *isw-1/cco-1* RNAi to evaluate mitochondrial stress and HSF-1::HA worms treated with EV, EV/*his-3* or *isw-1/his-3* RNAi to test histone stress. *isw-1* RNAi does not affect HSF-1 binding to its target genes either in conditions with histone- or mitochondrial stress (Supplementary Figs 3a-c). HSF-1 binding to the telomerase promoter (*trt-1*) is shown as a negative control, as it does not contain HSF-1 binding sites. Although these results exclude that ISW-1 interferes with HSF-1 binding to the promoter region of the sHSPs genes, we cannot exclude the possibility that ISW-1 regulates HSF-1 function through some other way.

7. The authors suggest an order of events, in which mitochondrial stress causes changes in histone expression, which leads to an increase in ISW-1 expression, which eventually triggers sHSP expression and extends lifespan. To properly support this order of events, histone expression under *cco-1* RNAi should be checked in *isw-1* mutant conditions. If the model is correct, then *isw-1* mutation should not affect histone expression.

We suppose that reviewer is asking whether histone expression is affected also in *isw-1* mutants during mitochondrial perturbation, so to ascertain that the dampened downstream effects (sHSP expression, lifespan extension) in *isw-1(n3297)* mutants are simply not due to the absence of changes in histone expression. To address this point, we examined histone expression in *isw-1(n3297)* mutants by qRT-PCR and Western blot. In the qRT-PCR experiment, *isw-1(n3297)* mutants show increased histone expression upon *cco-1* RNAi (Supplementary Fig 5b). *cco-1* RNAi treatment modulates histone H3 protein content in *isw-1(n3297)* mutants hence in a likewise manner as in N2 worms (Fig. 1a, Supplementary Fig 5c). These data demonstrate that the blunted downstream effects in *isw-1(n3297)* mutants upon *cco-1* RNAi treatment are not due to the lack of changes in histone expression.

8. As a model of mitochondrial ETC perturbation, the authors exclusively use *cco-1* RNAi. To make sure that the findings are not *cco-1* RNAi specific, the authors should validate their finding of an *isw-1*-dependent longevity in a different mitochondrial perturbation model, e.g. in *isp-1* mutants.

We performed a lifespan experiment by using the ETC complex I mutant *nuo-6(qm200)* worms with *isw-1* RNAi. Although the longevity phenotype of our *nuo-6(qm200)* mutants was not very strong, *isw-1* RNAi abolished their lifespan extension (Supplementary Fig. 5e), indicating that ISW-1 is not only specific to *cco-1* RNAi-mediated longevity.

Minor Comments:

9. Given that also other chromatin remodelers are upregulated in response to histone and mitochondrial stress, and some of them may affect lifespan, the authors should discuss which is the contribution of those other remodelers beside ISW-1 in the response to *cco-1* RNAi and *his-3* RNAi, as they did not go further in studying them.

We totally agree with the reviewer regarding the possibility that many chromatin remodeling factors contribute to the longevity deriving from mitochondrial- and histone perturbations. We highlight this point now in the revised manuscript (line 358).

10. There are several other long-lived *C. elegans* models, whose longevity depends on the actions of sHSPs. The authors should at least discuss, if they think that *isw-1* is involved in these other models.

We discuss the published data regarding the role of ISW-1 in the longevity of the long-lived *daf-2* insulin/IGF-1 signaling (IIS) mutants⁷ (line 334). We have also added a sentence highlighting the role of sHSPs in the longevity of IIS mutants (line 340). This section of the discussion combines the roles of ISW-1 with sHSPs and longevity, also in other long-lived *C. elegans* models.

11. The third section of the “results” section is very long compared with others. I would recommend to split it into two parts for better readability.

We agree with the reviewer, and have now divided the “Results” section into shorter paragraphs.

References

- 1 Durieux, J., Wolff, S. & Dillin, A. The cell-non-autonomous nature of electron transport chain-mediated longevity. *Cell* **144**, 79-91, doi:10.1016/j.cell.2010.12.016 (2011).
- 2 Galdieri, L., Zhang, T., Rogerson, D. & Vancura, A. Reduced Histone Expression or a Defect in Chromatin Assembly Induces Respiration. *Molecular and cellular biology* **36**, 1064-1077, doi:10.1128/MCB.00770-15 (2016).
- 3 Dang, W. *et al.* Inactivation of yeast Isw2 chromatin remodeling enzyme mimics longevity effect of calorie restriction via induction of genotoxic stress response. *Cell metabolism* **19**, 952-966, doi:10.1016/j.cmet.2014.04.004 (2014).
- 4 Kim, H. E. *et al.* Lipid Biosynthesis Coordinates a Mitochondrial-to-Cytosolic Stress Response. *Cell* **166**, 1539-1552 e1516, doi:10.1016/j.cell.2016.08.027 (2016).
- 5 Pellegrino, M. W. *et al.* Mitochondrial UPR-regulated innate immunity provides resistance to pathogen infection. *Nature* **516**, 414-417, doi:10.1038/nature13818 (2014).
- 6 Pothof, J. *et al.* Identification of genes that protect the *C. elegans* genome against mutations by genome-wide RNAi. *Genes & development* **17**, 443-448, doi:10.1101/gad.1060703 (2003).
- 7 Curran, S. P., Wu, X., Riedel, C. G. & Ruvkun, G. A soma-to-germline transformation in long-lived *Caenorhabditis elegans* mutants. *Nature* **459**, 1079-1084, doi:10.1038/nature08106 (2009).

Reviewers' Comments:

Reviewer #1:

Remarks to the Author:

The authors have satisfactorily addressed my previous concerns.

Reviewer #2:

Remarks to the Author:

This new version of the manuscript presented by Matilainen and colleagues shows a good effort, replying to all the comments made by the reviewers and including new data, some of which further supports the hypotheses presented in this study.

Regarding the response to my introduction:

The response by the authors has significantly clarified things to me and they have argued their case well.

Regarding my major and minor comments:

1) The new mRNA-seq data provides now a better view of the overall gene expression changes caused by *cco-1* and *his-3* RNAi. While it is clear that the two perturbations give very different responses overall, it actually looks like the overlap between their regulated genes is significant; and this possibility, in addition to the found co-regulation of histones and chromatin remodelers, actually helps the paper. The authors should generate a p-value for the overlap of the upper two circles of the Venn diagram and also state its significance in the text (assuming that it is significant). Further, the authors could consider to remove the *cco-1*RNAi vs *his-3*RNAi circle. I am not sure, if it actually adds value to the figure.

It is unfortunate that in the co-regulated genes only an enrichment for innate immunity go-terms as a go-term related to stress response and lifespan regulation could be reported by the authors. One would also hope for other stress response-related go-terms and chromatin-related terms to emerge. If the authors approach their analysis differently, starting from lists of chromatin factors, heat-shock response proteins, etc., are these lists significantly enriched amongst co-regulated genes? They should be. Please test.

Finally, Figures 2a and S1C were hard to read. Please change for better quality or make text larger.

2) OK. These *his-3*RNAi lifespan phenotypes are weak, but they are consistent with the story.

3) OK. Although, the authors should state somewhere in the manuscript or figure legend that the RT-qPCR data, is reproducible in different independent experiments, to avoid any confusion given the mild changes sometimes observed.

4) OK.

5) Due to the authors' difficulties with the *hil-1* RNAi clone, my main concern could not be addressed. Also the argument that *isw-1* OE did not alter *hil-1* expression only shows that *hil-1* is not functioning downstream of *isw-1*. But from my understanding of the story, *hil-1* should anyway act upstream of *isw-1* and therefore not get affected by *isw-1* OE.

6) MOSTLY OK. Thanks for testing at least this part of the potential mechanism.

7) OK.

8) OK, although the *nuo-6* lifespan phenotype here is indeed not looking strong. You might consider omitting this data from the manuscript.

9) OK.

10) OK.

11) OK.

Unfortunately, one new point emerged for me:

- The story predicts for me that *hsf-1* mutation should suppress the longevity of *cco-1* RNAi. Is this the case? If this is published knowledge, then please state and cite this. Otherwise I suggest to still conduct this experiment. Alternatively, the authors could test, if RNAi against the sHSPs suppresses *cco-1* RNAi longevity. Any of these two experiments would nicely support the implication from this story that the longevity caused by mitochondrial impairment is dependent on the expression of (s)HSPs through HSF-1.

New suggestions for minor changes.

- Line 74/75: I would delete "in *C. elegans*", unless the statement becomes then incorrect. Instead replace "worms" by "*C. elegans*" in line 75.

- Line 87 should end on a ",".

- In Line 179 I suggest to write "...Notably, also RNAi against..."

- Figure 3f: The major increase under NaCl is for histone H2A, which is different than for *cco-1* or *his-3* RNAi. This should be noted in the text. Further, the possibility that different stresses might deregulate different histones and still eventually function through a common downstream mechanism involving ISW-1 should be picked up in the discussion.

- The title "Changes in histone levels increase the expression of chromatin remodelling factor *isw-1*" in the Results section, I would consider replacing it for something different, as this observation has been already reported in the previous section.

Reviewer's comments:

We thank you again for your effort in reading and commenting on this manuscript. Based on your new comments, we have further modified our manuscript to make it stronger candidate for *Nature Communications*.

Reviewer #1

The authors have satisfactorily addressed my previous concerns.

We are happy to learn that we were able to address all concerns made by Reviewer #1.

Reviewer #1

This new version of the manuscript presented by Matilainen and colleagues shows a good effort, replying to all the comments made by the reviewers and including new data, some of which further supports the hypotheses presented in this study.

Regarding the response to my introduction:

The response by the authors has significantly clarified things to me and they have argued their case well.

We are happy to hear that Reviewer #2 is pleased with our revision.

Regarding my major and minor comments:

1) The new mRNA-seq data provides now a better view of the overall gene expression changes caused by *cco-1* and *his-3* RNAi. While it is clear that the two perturbations give very different responses overall, it actually looks like the overlap between their regulated genes is significant; and this possibility, in addition to the found co-regulation of histones and chromatin remodelers, actually helps the paper. The authors should generate a p-value for the overlap of the upper two circles of the Venn diagram and also state its significance in the text (assuming that it is significant). Further, the authors could consider to remove the *cco-1*RNAi vs *his-3*RNAi circle. I am not sure, if it actually adds value to the figure.

It is unfortunate that in the co-regulated genes only an enrichment for innate immunity go-terms as a go-term related to stress response and lifespan regulation could be reported by the authors. One would also hope for other stress response-related go-terms and chromatin-related terms to emerge. If the authors approach their analysis differently, starting from lists of chromatin factors, heat-shock response proteins, etc., are these lists significantly enriched amongst co-regulated genes? They should be. Please test.

Finally, Figures 2a and S1C were hard to read. Please change for better quality or make text larger.

We appreciate the Reviewer's insight on how to improve the presentation of our RNA-seq data. We split the Venn diagram into two parts (up- and downregulated genes), and found that the overlap is significant between *cco-1* and *his-3* RNAi treatments in both diagrams (Supplementary Fig. 1b). We also removed the *cco-1* RNAi vs *his-3* RNAi circle (Supplementary Fig. 1b). To highlight the commonly upregulated chromatin remodeling-

and heat shock response, we created custom gene sets (HSP40, HSP60, HSP70 and HSP90 class heat shock proteins; Small heat shock proteins; Chromatin remodelers) (Supplementary Table 1). Importantly, these custom gene sets were found to be significantly enriched in GSEA (Supplementary Fig. 1c). We also improved Fig. 2a and remade Supplementary Fig. 2c to improve readability.

2) OK. These *his-3* RNAi lifespan phenotypes are weak, but they are consistent with the story.

We agree that the *his-3* RNAi lifespan phenotypes are weak. However, they are consistent between repeated experiments.

3) OK. Although, the authors should state somewhere in the manuscript or figure legend that the RT-qPCR data, is reproducible in different independent experiments, to avoid any confusion given the mild changes sometimes observed.

We added sentence to the manuscript to highlight the reproducibility (page 7, first paragraph).

4) OK.

5) Due to the authors' difficulties with the *hil-1* RNAi clone, my main concern could not be addressed. Also the argument that *isw-1* OE did not alter *hil-1* expression only shows that *hil-1* is not functioning downstream of *isw-1*. But from my understanding of the story, *hil-1* should anyway act upstream of *isw-1* and therefore not get affected by *isw-1* OE.

We agree that the potential role of HIL-1 in responses against mitochondrial- and histone stress could not be addressed. However, since the role of ISW-1 in mitochondrial- and histone stress responses is the main message of this manuscript, we feel that further examination of HIL-1 in this context is out of the scope of this paper.

6) MOSTLY OK. Thanks for testing at least this part of the potential mechanism.

7) OK.

8) OK, although the *nuo-6* lifespan phenotype here is indeed not looking strong. You might consider omitting this data from the manuscript.

We agree with the reviewer, and removed *nuo-6* lifespan from the manuscript.

9) OK.

10) OK.

11) OK.

Unfortunately, one new point emerged for me:

- The story predicts for me that *hsf-1* mutation should suppress the longevity of *cco-1* RNAi. Is this the case? If this is published knowledge, then please state and cite this. Otherwise I suggest to still conduct this experiment. Alternatively, the authors could test, if RNAi against the sHSPs suppresses *cco-1* RNAi longevity. Any of these two experiments would nicely support the implication from this story that the longevity caused by mitochondrial impairment is dependent on the expression of (s)HSPs through HSF-1.

We addressed this point in the first revision, and based on Reviewer #1's suggestion we examined if RNAi against sHSPs suppresses *cco-1* RNAi-mediated longevity. We found that combined sHSP RNAi does not shorten the lifespan of *cco-1* RNAi treated worms (Reply to Reviewers Fig. 1). Notably, when shortening *isw-1* OE strain lifespan with sHSP knockdown (combined *hsp-16.1*, *hsp-16.2*, *hsp-16.41* and *hsp-16.48* RNAi), each sHSP RNAi was diluted to 1/4 of the total RNAi mixture (see Fig. 5d). In simultaneous knockdown with *cco-1*, each sHSP RNAi bacteria was diluted to 1/8 of the total RNAi mixture (*cco-1* RNAi was 1/2), and therefore, sHSP knockdown efficiency may not have been sufficient. Due to this concern, we are reluctant to draw conclusions from this experiment, and in our opinion, these data should not be included to the manuscript (see also out previous reply to reviewers, point 4 from the reviewer #1).

Reply to Reviewers Fig. 1. Lifespan of *cco-1/hsp-16.1, hsp-16.2, hsp-16.41, hsp-16.48* RNAi treated worms compared to that of EV/*cco-1* RNAi treated worms. Experiment was done twice with similar results.

New suggestions for minor changes.

- Line 74/75: I would delete "in *C. elegans*", unless the statement becomes then incorrect. Instead replace "worms" by "*C. elegans*" in line 75.

We have modified the manuscript accordingly.

- Line 87 should end on a ",".

We have modified the manuscript accordingly.

- In Line 179 I suggest to write "...Notably, also RNAi against..."

We have modified the manuscript accordingly.

- Figure 3f: The major increase under NaCl is for histone H2A, which is different than for *cco-1* or *his-3* RNAi. This should be noted in the text. Further, the possibility that different

stresses might deregulate different histones and still eventually function through a common downstream mechanism involving ISW-1 should be picked up in the discussion.

We added sentence stating that *isw-1* expression is increased by stressors that modulate histone levels, although the histone levels are not affected in identical manner (page 8, end of the 2nd paragraph). Additionally, we modified the discussion accordingly (page 14, third paragraph, page 15, first paragraph).

- The title “Changes in histone levels increase the expression of chromatin remodelling factor *isw-1*” in the Results section, I would consider replacing it for something different, as this observation has been already reported in the previous section.

We have changed the title to “Different stressors modulate *isw-1* expression”.